# Vortex phase matching as a strategy for schooling in robots and in fish

Liang Li [1,2,3,4], Máté Nagy [1,2,3,5,6,7 ✉], Jacob M. Graving [1,2,3], Joseph Bak-Coleman [8], Guangming Xie [4,9,10 ✉] & Iain D. Couzin [1,2,3 ✉]

It has long been proposed that flying and swimming animals could exploit neighbour-induced flows. Despite this it is still not clear whether, and if so how, schooling fish coordinate their movement to benefit from the vortices shed by others. To address this we developed bio-mimetic fish-like robots which allow us to measure directly the energy consumption associated with swimming together in pairs (the most common natural configuration in schooling fish). We find that followers, in any relative position to a near-neighbour, could obtain hydrodynamic benefits if they exhibit a tailbeat phase difference that varies linearly with front-back distance, a strategy we term 'vortex phase matching'. Experiments with pairs of freely-swimming fish reveal that followers exhibit this strategy, and that doing so requires neither a functioning visual nor lateral line system. Our results are consistent with the hypothesis that fish typically, but not exclusively, use vortex phase matching to save energy.

[1] Department of Collective Behaviour, Max Planck Institute of Animal Behavior, Konstanz, Germany. [2] Centre for the Advanced Study of Collective Behaviour, University of Konstanz, Konstanz, Germany. [3] Department of Biology, University of Konstanz, Konstanz, Germany. [4] State Key Laboratory for Turbulence and Complex Systems, and Intelligent Biomimetic Design Lab, College of Engineering, Peking University, Beijing, P. R. China. [5] MTA-ELTE Statistical and Biological Physics Research Group, Hungarian Academy of Sciences, Budapest, Hungary. [6] MTA-ELTE "Lendület" Collective Behaviour Research Group, Hungarian Academy of Sciences, Budapest, Hungary. [7] Department of Biological Physics, Eötvös Loránd University, Budapest, Hungary. [8] Department of Ecology and Evolutionary Biology, Princeton University, Princeton, NJ, USA. [9] Institute of Ocean Research, Peking University, Beijing, P. R. China. [10] Peng Cheng Laboratory, Shenzhen, P. R. China. ✉email: nagymate@hal.elte.hu; xiegming@pku.edu.cn; icouzin@ab.mpg.de

It has long been hypothesised that schooling fish may be able to obtain hydrodynamic benefits by swimming closely with others[1–3]. These benefits are possible because the propulsive motion of swimming fish creates regular alternating patterns of vortices in water which have the potential to be exploited by conspecifics, allowing them to save energy[4] or to generate thrust[5]. A simple two-dimensional model of this process led Weihs to propose that schooling fish in energetically demanding environments would be expected to adopt a diamond lattice structure to exploit the vortices produced by those ahead[3]. While compelling, Partridge and Pitcher subsequently found no evidence for such a strict regulation of relative spatial positioning in four species of fish[6], and neither did other studies of further species[7,8]. More recent work has suggested that, in addition to spatial considerations, fish likely need to adjust the timing of their tailbeat relative to the movement of their neighbours (phase difference between the undulation patterns of the two fish)[9–12], but to date, and with an abundance of theoretical work describing the potential for hydrodynamic benefits in fish schools[3,9,11], we still lack a coherent biologically plausible mechanism for what to expect of individuals' behaviour if they have, indeed, evolved mechanisms to exploit the vortices produced by others. Furthermore, experimental verification of the potential mechanisms in real fish schools has remained elusive[6].

Due to the hypothesis that energy savings may be possible when swimming together, several attempts have been made to simply show reduced energy consumption in collective contexts[8,13,14], often relying on comparing per-capita metabolic output as a function of group size[8,14]. This approach, while suggestive, inherently confounds the physiological effects of stress, and thus metabolic rates; individuals' stress typically reduces with increasing group size resulting in collective energy savings unrelated to hydrodynamics[15]. Furthermore, these studies provide limited mechanistic insight into how fish reduce energy expenditure. A complementary approach involves measuring tailbeat frequency[8,13], but this faces similar confounds with respect to levels of fear[16].

Because of the complexity of these inter-relationships, it is challenging to determine the relative contributions of social stress and energy savings in these contexts. In an attempt to make progress with this problem, researchers have proposed numerical[9,11,17,18] and simplified physical models[19–24], and a large number of hypotheses regarding hydrodynamic benefits related to spatial differences and/or phase differences have been generated[3,9,11,12,17–32]. Broadly speaking, these studies have proposed that benefits may arise from fish occupying specific spatial positions with respect to neighbours within groups and/ or that fish modulate the phase of their tailbeat with respect to others in order to obtain energetic benefits.

It is clear that, in nature, fish do not typically adopt specific spatial positions relative to one another—which among the hypotheses is more or less the only one that is easy to evaluate[3,6]. Due to the diversity of other hypotheses, and a lack of knowing what to look for in behavioural data (e.g., reinforcement learning[11] predicts that fish could achieve hydrodynamic benefits but it is unclear how they may actually do so, behaviourally), and due to the considerable difficulty associated with measuring the detailed flow fields around the fish's body, we lack systematic experimental (biological) validation of the many hypotheses that have been generated regarding hydrodynamic interactions among fish. In summary, while there exists an abundance of hypotheses for what real fish may do to gain hydrodynamic benefits, we presently lack experimental evidence to show whether fish exploit neighbour-generated hydrodynamics when schooling, or if they do how they do so.

Our focus here is to develop a unified understanding of the previously described hypotheses regarding how fish should be expected to behave if they are exploiting hydrodynamic interactions (i.e. to reveal the behavioural rule(s) that fish would be expected to adopt to achieve benefits), and test our predictions in experiments conducted with real fish.

Since direct measurement of the hydrodynamic costs of swimming together in real fish is not possible, and since evaluating costs with numerical or traditional physical models is difficult at high (biologically realistic) Reynolds numbers ($o(10^5)$), we first develop, and utilise, bio-mimetic fish-like robots with which we can directly measure the power consumption associated with swimming together. This approach offers several key benefits. Our bio-mimetic robots recreate important features of real fish with high fidelity, including a 3D body with a flexible caudal fin, and actively controlled body undulations. Therefore, the swimming motion and reverse Kármán vortices exhibited by real swimming fish are reproduced. Additionally, because the robots are controlled electronically, we can directly measure the energetic costs of locomotion for individuals in different spatial positions and dynamical regimes. Finally, we can also easily visualise the fluid flow during interactions to gain insight into the underlying energy saving mechanisms. This approach, combined with the development of a minimal model of the core hydrodynamic mechanisms at work, allows us to reveal a biologically plausible rule that, if adopted by following fish, would allow them to obtain hydrodynamic benefits from a near neighbour irrespective of their exact relative spatial position. Automated tracking and body-posture analysis of pairs of freely swimming fish (goldfish, *Carassius auratus*) reveal that followers do adopt this rule, and that it typically allows them to save energy. Furthermore, by conducting experiments in which we reversibly impair the sensing capabilities of fish, we find that doing so requires neither a functioning visual or lateral line system. The simplicity and robustness of this rule suggests that it may be widespread, and that it could also serve as a design principle for improving the efficiency of fish-like underwater vehicles.

## Results

**Experiments with robotic fish**. We developed, and employed, a bio-mimetic robotic fish platform (Fig. 1, Supplementary Figs. 1–4 and Movie 1 and 2) in order to experimentally evaluate the costs and benefits of swimming together. We constructed two identical robotic fish, 45 cm in length and 800 g in mass. Each has three sequential servo-motors controlling corresponding joints, covered in a soft, waterproof, rubber skin. In addition, the stiffness of the rubber caudal fin decreases towards the tip[33] (Supplementary Fig. 1). The motion of the servomotors is controlled using a bio-inspired controller called a central pattern generator (CPG)[34,35] resulting in the kinematics that mimic normal real fish body undulations when swimming[36] (see Supplementary Fig. 2 and Note 1). Here, due to the complexity of the problem (as discussed above) we consider hydrodynamic interactions between pairs of fish. We note that this is biologically meaningful as swimming in pairs is both the most common configuration found in natural fish populations[7,10,37,38], and it has been found that even in schools fish tend to swim close to only a single neighbour[7,37].

To evaluate the energetics of swimming together we conducted experiments on our pair of robotic fish in a flow tank (test area: 0.4-m-wide, 1-m-long and 0.45-m-deep; Fig. 1b and Supplementary Fig. 3). In order to conduct such an assessment we first measured the speed of our robots when freely swimming alone (we did so in a large tank 2-m-wide, 3-m-long and 0.4-m-deep). We then set the flow speed within our flow tank to this

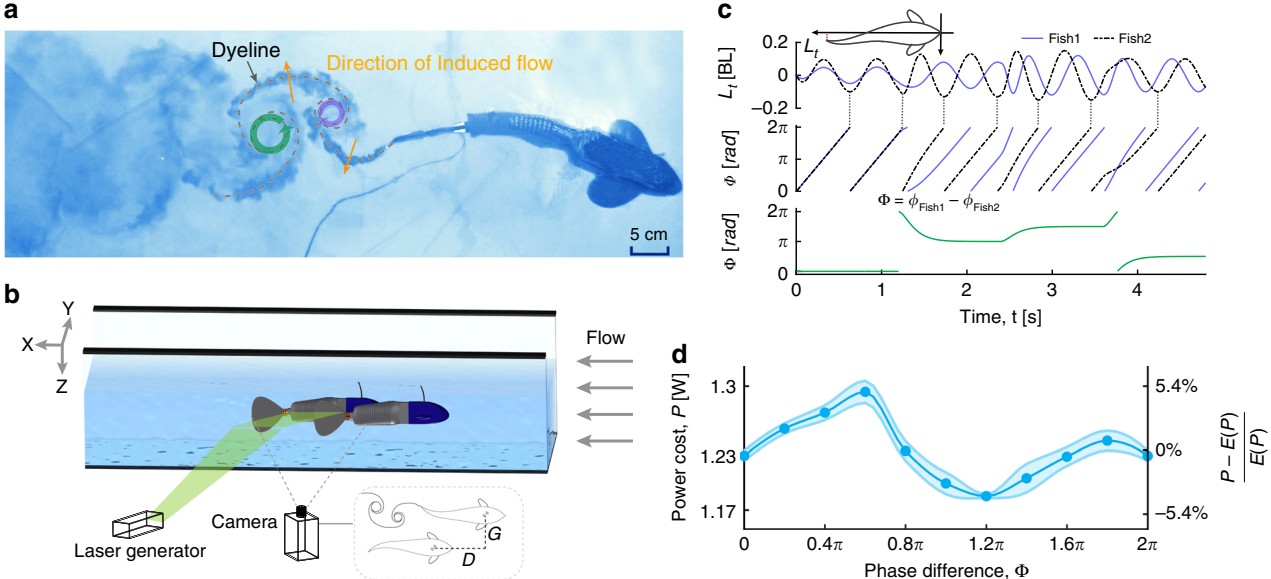

**Fig. 1 The robotic fish platform employed to investigate hydrodynamic benefits of schooling. a** The reverse Karman vortices shedding by the robotic fish with dye flow visualisation. **b** Schematic view of the setup that allows setting various spatiotemporal differences between two robotic fish swimming in a flow tank (Front-back distance $D \in [0.22, 1]$ BL (body length), Left-right distance $G \in [0.27, 0.33]$ BL and Phase difference $\Phi \in [0, 2\pi]$). A laser generator was used to visualise the hydrodynamic interactions (see Supplementary Note 1). **c** The phase difference $\Phi$ is evaluated by the difference between the undulation phase of the two robots. Undulation phase is evaluated based on the lateral position $L_t$ of the tail tip. **d** Power cost (absolute value on the left $y$-axis, and relative value compared to the average power cost on the right $y$-axis), is shown as a function of the phase difference at $D = 0.33$ BL and $G = 0.27$ BL. Error bars are standard error of the mean.

free-swimming speed ($0.245\,\mathrm{ms}^{-1}$) allowing us to ensure the conditions in the flow tank are similar to those of the free-swimming robot. Unlike in the solitary free-swimming condition, to have precise control of spatial relationships in the flow tank we suspended each robotic fish by attaching a thin aluminium vertical bar to the back of each robot, which was then attached to a step motor above the flow tank (Supplementary Fig. 3 and Movie 2). To establish whether the robotic fish connected with a thin bar has similar hydrodynamics compared to when free swimming, we measured the net force (of the drag and thrust generated by the fish body in the front-back direction) acting on the robot in the flow tank. The measured net force over a full cycle (body undulation) was found to be zero; thus the bar is not measurably impacting the hydrodynamics of our robot fish in the front-back direction as they swim in the flow tank (Supplementary Fig. 5).

To further validate the utility of the platform, we also compared the power consumption of our robots swimming side-by-side, for different relative phase differences $\Phi$, with equivalent measurements made with a simple 2D computational fluid dynamics (CFD) model of the same scenario (Supplementary Note 2). In both cases (see Supplementary Fig. 6a, c for robotic experiments and CFD simulations, respectively) we find that there exists an approximately sinusoidal relationship between power costs and phase difference which is defined as $\Phi = \phi_{\mathrm{leader}} - \phi_{\mathrm{follower}}$ (Fig. 1c, d). Due to the 2D nature of the simulation, as well as many other inevitable differences between simulations and real world mechanics, the absolute power costs are different from those measured for the robots, but nevertheless the results from these two approaches are broadly comparable and produce qualitatively similar relative power distributions when varying the phase difference between the leader and follower. These results indicate that our robotic fish are both an efficient (making estimates of swimming costs is far quicker with our robotic platform than it is with CFD simulations) and effective (in that they capture the essential hydrodynamic interactions as well as

naturally incorporate 3D factors) platform for generating testable hypotheses regarding hydrodynamic interactions in pairs of fish.

We subsequently utilise our robots to directly measure the energy costs associated with swimming together as a function of relative position (front-back distance $D$ from 0.22 to 1 body length (BL) in increments of 0.022 BL and left-right distance $G$ from 0.27 to 0.33 BL in increments of 0.022 BL) while also varying the phase relationships (phase difference, $\Phi$) of the body undulations exhibited by the robots (the phase of the follower's tailbeat $\phi_{\mathrm{follower}}$ relative to that of the leader's $\phi_{\mathrm{leader}}$, Fig. 1c).

By conducting 10,080 trials (~120 h of data), we obtain a detailed mapping of the power costs relative to swimming alone associated with these factors (Fig. 2a). Such a mapping allows us to predict how real fish, that continuously change relative positions[6,8], should correspondingly continuously adjust their phase relationship in order to maintain hydrodynamic benefits. To quantify the costs we determine the energy required to undulate the tail of each robot allowing us to define, and calculate, a dimensionless relative power coefficient as:

$$\eta = \frac{(P_1^{\mathrm{Water}} - P^{\mathrm{Air}}) - (P_2^{\mathrm{Water}} - P^{\mathrm{Air}})}{P_1^{\mathrm{Water}} - P^{\mathrm{Air}}} = \frac{P_1^{\mathrm{Water}} - P_2^{\mathrm{Water}}}{P_1^{\mathrm{Water}} - P^{\mathrm{Air}}}, \quad (1)$$

where $\eta$ is the relative power coefficient, $P^{\mathrm{Air}}$, $P_1^{\mathrm{Water}}$ and $P_2^{\mathrm{Water}}$ are the power costs of the robotic fish swimming in the air (an approximation of the dissipated power cost due to mechanical friction, resistance, etc. within the robot that are not related to interacting with the water), alone in water, or in a paired context in the water, respectively. $P_1^{\mathrm{Water}} - P^{\mathrm{Air}}$ and $P_2^{\mathrm{Water}} - P^{\mathrm{Air}}$ therefore represent the power costs due to hydrodynamics while swimming alone, and in a pair, respectively (see Methods section). Correspondingly, the coefficient $\eta$ compares the energy cost of fish swimming in pairs to swimming alone. Positive values (blue in Fig. 2a) and negative values (red in Fig. 2a) respectively represent energy saving and energy cost relative to swimming

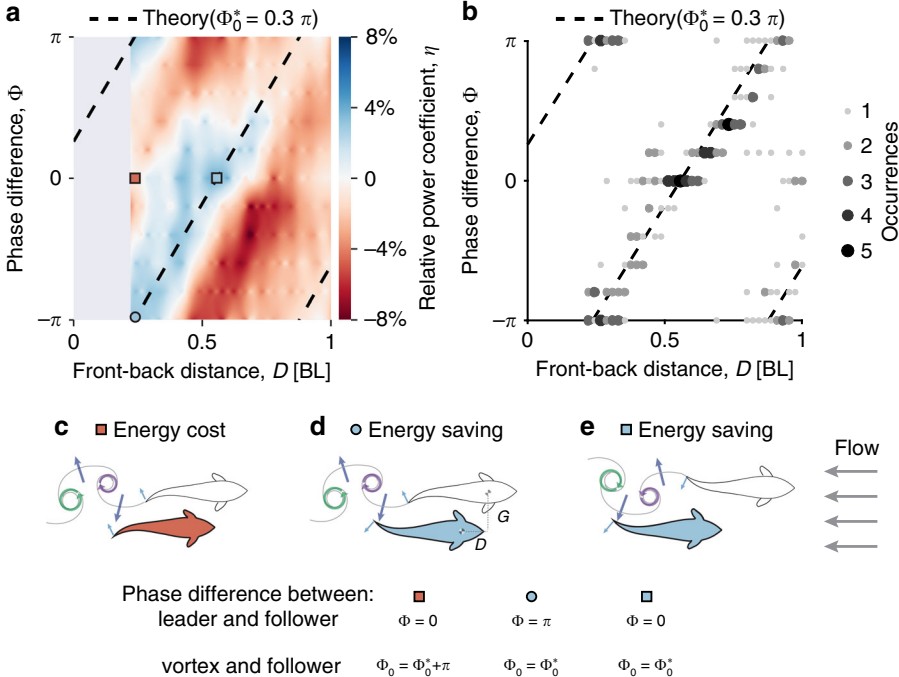

**Fig. 2 Robotic fish save energy by vortex phase matching (VPM). a** Relative power coefficient $\eta$ shown as a function of the phase difference between the leader and the follower $\Phi$ and front-back distance $D$ at left–right distance $G = 0.31$ BL. The dashed line (also in **b**) shows the functional relationship described in Eq. (2) that determines the theoretical phase relationship that maximally saves energy (Methods section). $\Phi_0^*$ is the optimal initial phase difference (fitted to the data points of maximum energy saving, as shown in **b**). The points marked by red square, blue circle and blue square indicate example cases depicted on panels **c**–**e**. **b** Location of maximal energy saving in the robotic trials. Point size and darkness denote the number of occurrences of each phase difference value at each front-back distance. **c**–**e** An illustration of important spatial configurations for vortex phase matching. Energy cost is related to how the follower moves its body relative to the direction of the induced flow of the vortices, in the opposite direction with $\Phi_0 = \Phi_0^* + \pi$ (**c**) or in the same direction with $\Phi_0 = \Phi_0^*$ (**d**, **e**). Followers interact with the induced flow of vortices with the same body phase at any front-back distance (within the range of hydrodynamic interactions), termed vortex phase matching. (**d**, **e**; $\Phi_0 = \Phi_0^*$ describes the hydrodynamic interaction resulting in energy saving, see description in the text). As the front-back distance changes, the followers must dynamically adopt phase difference $\Phi$, with respect to that of the leader.

alone. The difference between the maximum energy saving and maximum energy cost for the robots is ~13.4%.

Our results indicate that there exists a relatively simple linear relationship between front-back distance and relative phase difference of the follower that minimises the power cost of swimming (as indicated by the dashed lines in Fig. 2a, b, the theoretical basis of which we will discuss below). This suggests that a follower can minimise energetic expenditure (and avoid substantial possible energetic costs) by continuously adopting a unique phase difference $\Phi$ that varies linearly as a function of front-back distance $D$ (see Fig. 2b for example), even as that distance changes. We find that while left-right distance $G$ does alter energy expenditure, this effect is minimal when compared to front-back positioning, and has little effect on the above relationship (Supplementary Figs. 7 and 8) in the range explored here.

Although we know fish generate reverse Kármán vortices at the Reynolds number (Re $= Lu/v \approx 10^5$, where $L$ is the fish body length, $u$ is the swimming or flow speed and $v$ is the kinematic viscosity) in our experiments[39] (Supplementary Fig. 9 and Movie 1), turbulence will dominate over longer distances[18]. In accordance with this, we see a relatively fast decay in the benefits of swimming together as a function of $D$ (e.g., $D > 0.7$ BL, Supplementary Fig. 10), a feature we also expect to be apparent in natural fish schools (where it would likely be exacerbated by what would almost always be less-laminar flow conditions). Therefore, we expect, based on our results, that hydrodynamic interactions are dominated by short-distance vortex-body interactions (with

$D < 0.7$ BL). While complementary to previous studies of vortex–vortex interactions[3,17,26], here we focus on vortex-body interactions since these are expected, based on the above results, to have a far greater impact on energetics or thrust when swimming (for this regime of Reynolds number).

**A simple mechanistic explanation and testable predictions.** In order to gain insights into why we see the results we do in our robotic experiments, and to develop experimentally testable predictions, we both visualised the fluid flow generated by our robots and we developed a deliberately simple hydrodynamic model that can account for our experimental findings.

Visualisations of the shed vortices downstream of the robotic fish (estimated by visualising the motion of small hydrogen bubbles introduced into the flow) reveal that in the energetically beneficial regions (blue colour in Fig. 2a) the direction of the follower's tail during the moving coincides with the direction of the induced flow of the vortices shed by the leader (Fig. 2d, e, Supplementary Fig. 11 and Movie 3). Thus energy savings seem predominantly determined by macroscopic properties of the flow, and specifically the follower's reaction to the primary induced flow of the coupled reverse Kármán vortex (Fig. 2c–e and Supplementary Fig. 11) produced by the leader.

To relate these findings to the behaviour of real fish (if present in real fish schools we would expect to only find features that are robust to natural sources of environmental and sensory/decision-making/motor noise) we derived a simple analytic model,

inspired by previous studies[12,28,40,41], and based on simplified hydrodynamic factors, from which we find we can describe the phase difference $\Phi$ that, if adopted by a follower, would allow it to continuously and dynamically optimise its swimming phase (as relative position changes) to obtain hydrodynamic benefits, such as to minimise energetic expenditure (Supplementary Fig. 12 and Note 3). The main assumption is that the flow is inviscid, and thus the vortex moves backward with constant morphology and velocity. This is reasonable in our case since, guided by our robotic results, we focus on the direction of the induced flow of the reverse Kármán vortices and short-range hydrodynamic interactions between the vortex and the body. Under this assumption, phase difference $\Phi$ is predicted to relate to the front-back distance $D$, as well as the leader's tailbeat frequency $f$, and swim speed $u$, in the following way:

$$\Phi = \frac{2\pi f}{u}D + \Phi_0 \qquad (2)$$

where $\Phi_0$ characterises the nature of the hydrodynamic interaction; the phase difference between the undulation of the follower fish and the vortex induced flow that it interacts with. Since $\Phi_0$ also describes the phase difference at $D = 0$, we term this the initial phase difference. In practice, when estimating this value from our experimental data (be it from robots as here, of for real fish as below) we must estimate the relationship between $\Phi$ and $D$, employing the full range of $D$ to best assess the phase difference, $\Phi$, at $D = 0$. This experimentally fitted $\Phi_0$ is denoted as $\Phi_0^*$. This is necessary for each experimental system due to inherent differences that exist between them, such as in body size, body morphology, body flexibility, differences in surface friction, and so on. If Eq. (2) with a specific $\Phi_0^*$ predicts phase difference $\Phi$ for a range of $D$, it indicates that the follower maintains a specific type of hydrodynamic interaction with the induced flow of the shed vortices across this range. However, the model alone can not tell if this specific interaction results in energy saving or not, since it depends on the value of $\Phi_0^*$ and, as noted above, on the details of the specific system. Thus the actual value of $\Phi_0^*$ for different systems (e.g. robotic and real fish) are not directly comparable. In the specific case of the robotic system, where the fitting can be made directly from values of energy consumption at different relative phases, $\Phi_0^*$ corresponds to maximum energy saving, and therefore $\Phi_0^* + \pi$ corresponds to maximum energetic costs.

The definition of the phase difference $\Phi$ in the rule (Eq. (2)) is analogous to the concept of spatial phase, with the first term corresponding to the spatial phase contribution (how much the phase is changed as a result of the travel of the vortex) and the second term (initial phase difference $\Phi_0$) corresponding to the schooling number $S$ in Becker et al.[28] (Supplementary Discussion). We decided to use $\Phi_0$ as it fits better the description of the biological system where each swimming characteristic is dynamically changing in time (such as the front-back distance $D$, undulation frequency $f$, etc.). As it remains possible that other swimming gaits and hydrodynamic interactions may allow for different energy saving mechanisms[3,10,42,43], we refer to this specific relationship, and the associated mechanism, as vortex phase matching, or VPM.

Despite its simplicity, our model accurately characterises the experimental relationship between front-back distance $D$, and phase difference $\Phi$, found in our robotic experiments (Fig. 2a, b). To evaluate the robustness and the validity of the rule (Eq. (2)), we further tested which relationship between frequency $f$ and the phase difference $\Phi$ results in a follower robot saving the most energy. We found this relationship is linear as a function of front-back distance and can also be described well by our simple rule with the slope of the linear function directly given by measured quantities (frequency $f$ and swim speed $u$) and the fitting only

needed for one constant (the intercept), the initial phase difference $\Phi_0 = \Phi_0^*$ (Supplementary Fig. 13). This further supports the view that our results can be accounted for by the main hydrodynamic features present, and that details, while likely refining this understanding, will not fundamentally change it. Furthermore this theory is consistent with all previous results (energy saving is related to both spatial and temporal differences in fish school[3,8,11,28]), providing a simple, unifying explanation of what may at first appear to be disparate results (Supplementary Discussion). In addition, and perhaps most importantly, it suggests that despite the complexity of fluid dynamics, and of real organisms, there may be a surprisingly simple rule (as evident by the simple linear relationship between phase difference $\Phi$ and front-back distance $D$), that if adopted by real fish, would allow them to continuously obtain hydrodynamic benefits from near neighbours, such as to minimise energetic costs. This is not to say that real fish would be expected to exhibit such a rule all the time —they face many other challenges such as to obtain food and avoid predators, as well as to move to gain and utilise social information[38,44,45]—but since this is both simple and robust, it opens up the possibility that it may be a previously undiscovered, but general, behavioural strategy.

**Testing our predictions with real fish.** To determine if fish exhibit hydrodynamic interactions, and specifically if they adopt vortex phase matching (VPM), as predicted by our rule (Eq. (2)), we conducted experiments on freely swimming pairs ($n = 32$ individuals) of goldfish in which fish were capable of sensing others with both their visual (V+) and lateral line (LL+) system.

To also gain insight into the role of different sensory modalities in regulating possible hydrodynamic interactions, we also conducted experiments with treatments where we (reversibly) impaired vision (V-LL+), the lateral line system (V+LL-) or both (V-LL-) (Supplementary Fig. 14). Fish were placed in a flow tank with uniform flow speed $u$, with $u$ ranging from 1.2 to 1.6 BLs$^{-1}$ (equivalent to their natural swimming speed[46]) with increments of 0.1 BLs$^{-1}$, and filmed with a camera at 100 frames-per-second (Supplementary Fig. 15 and Methods section) from which the body posture of each fish was estimated automatically using DeepPoseKit[47] (Fig. 3a and Supplementary Fig. 16 and Movie 4). We analysed all data where leader-follower pairs were spatially positioned such that the follower could potentially interact with the vortices shed by the leader (see above), up to a separation of 1 BL in front-back distance (801,000 frames; Supplementary Note 4). For each front-back distance, we show the distribution of the phase difference of the fish (each column in Fig. 3b, c). The peak of the distribution of the phase difference (which we get by fitting a circular mean) is linearly correlated to the front-back distance (the bright area, Fig. 3b, c).

To test whether the observed linear relationship results from VPM, we employed Eq. (2), for each moment in time, to determine the phase difference that is predicted to occur—if the fish were employing VPM with a fixed $\Phi_0$—across all front-back distances. We note that unlike for the robotic fish, we cannot conduct fitting directly to measured energetic costs, but can do so via measured body kinematics. Finding $\Phi_0^*$ in this case would indicate that, despite being a highly dynamic scenario—real fish constantly change their relative positions with respect to one another—they nonetheless adopt a consistent type of hydrodynamic interaction that is described by our model (Eq. (2), with the slope given directly from the measured quantities frequency $f$ and swim speed $u$). By comparing the predicted and observed phase difference over the full range of $D$ using a periodic least square regression algorithm (Methods section), we find the value of $\Phi_0^* = -0.2\pi$. Although the predicted relationship (Fig. 3d)

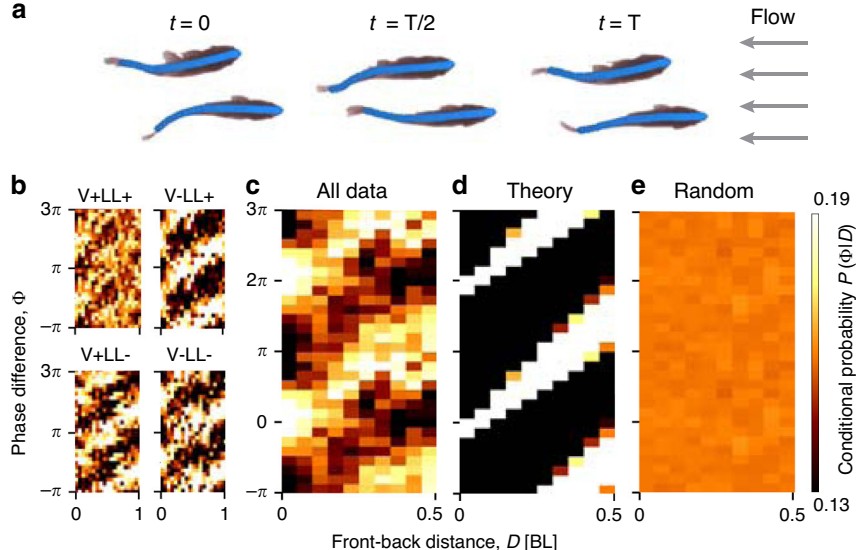

**Fig. 3 Real fish exhibit vortex phase matching (VPM). a** Example frames from a video recorded in the flow tank, overlaid with the result of deep-learning-based body posture tracking. The phase of the undulations of each fish was determined by the motion of the tail tip (measured at the caudal peduncle). **b–e** Occurrences of observed swimming relationships (probability density function) as a function of phase difference $\Phi$ and front-back distance $D$ between leader and follower illustrated by colour-coding for one case with intact vision and lateral line (V+LL+), three cases with impaired vision (V-LL+), the lateral line (V+LL-) or both (V-LL-) (**b**), and all pooled data (**c**; V+LL+: $n = 6$, V+LL-: $n = 5$, V-LL+: $n = 3$, V-LL: $n = 2$; $n$ given is the number of pairs). Data are duplicated twice along the phase difference axis to more clearly demonstrate the periodic pattern. Theoretical predictions (**d**) are derived by transforming observed swimming characteristics (including frequency $f$, front-back distance $D$ and flow speed $u$) using our model (Eq. (2)) with $\Phi_0 = \Phi_0^* = -0.2\pi$. $\Phi_0^*$ is determined by comparing experimental data and theoretical predictions after Von Mises fitting with a periodic least square fitting algorithm (Methods section). Randomised data (**e**) are generated by swapping two fish postures respectively from two randomly selected trials within same treatment (Supplementary Note 4).

lacks the noise present in the real biological system, visually it can be seen that the data (Fig. 3b, c) correspond to our predicted relationship (Fig. 3d and Supplementary Figs. 17–19), and randomisation tests (Fig. 3e and Supplementary Fig. 20) show that the probability of measuring the observed VPM pattern by chance is very low ($P = 0.002$; Supplementary Fig. 21). This holds true also for each of the four sensory treatments: intact, V+LL+; and three impaired, V-LL+, V+LL- and V-LL- (Supplementary Fig. 22), indicating that the VPM is robust and may be a result of passive hydrodynamic interactions[12,27,28] or predominantly a preflex, or proprioceptive[48–50] response to neighbour-generated hydrodynamic cues (see discussion below). This finding suggests that both the visual and lateral line systems may be free to process other valuable sources of information such as non neighbour-generated flow cues, possible threats in the environment and the movement of conspecifics (in relation to the latter, differences seen among the sensory treatments (Fig. 3b) are largely attributable to impaired fish swimming more closely together (Supplementary Fig. 23)).

While the above analysis demonstrates that fish are employing VPM, it is not possible by obtaining the $\Phi_0^*$ alone to determine why they are doing so. This is because $\Phi_0^*$ is fitted by estimating the typical phase difference at each front-back distance, and as discussed above, $\Phi_0^*$ depends on system specific characteristics such as body morphology, body size, and so on. Unlike in the robotic fish experiments, where we can directly measure power consumption, such measurement is not possible for the real fish. We cannot directly apply the measured costs and benefits of swimming together from our bio-mimetic robots, due to inherent differences between them such as propulsive undulations travelling through the body of real fish (which is flexible and elastic) more smoothly than in our robotic fish (that has only

three joints), and the skin of fish being coated with mucus to decrease resistance to fluids[51] etc. We note that as a result of these factors real fish are likely to obtain considerably greater benefits if they employ VPM than do our robots.

In order to gain insight into why fish perform VPM we conducted an additional analysis of the power consumption of the follower fish (by approximating it from the measured amplitude and frequency of its body undulation[48]) for different types of hydrodynamic interactions (the full range of $\Phi_0$ values, characterising the tail moving with, or against, the vortex induced flow for example). Since fish in the flow tank tend to match their swim speed to the flow speed of the water (see Supplementary Fig. 24), here we are not considering absolute energy savings (which would be maximised by not swimming at all), but rather the relative power consumption for a given swim speed in the presence, or absence, of hydrodynamic interactions.

To evaluate the body kinematics in the absence of hydrodynamic interactions we consider how frontal individuals swim when the other fish is far behind ($D > 2$ BL), and it thus cannot benefit from neighbour-generated vortices. We also chose this method since isolating the fish would likely induce stress responses that could confound our results. To evaluate body kinematics in the presence of vortices we analysed the body undulations of the follower when in close proximity (within 0.4 BL), where hydrodynamic effects will be strongest (Supplementary Fig. 25). We find that in the vicinity of vortices, fish exhibit a higher tailbeat amplitude and lower tailbeat frequency (Supplementary Fig. 26), which indicates less power consumption[48].

To further test if fish can save energy by adopting VPM with the typical vortex-body hydrodynamic interactions ($\Phi_0 = -0.2\pi$), we compared an estimation of the power consumption under different hydrodynamic interactions. Since the hydrodynamic

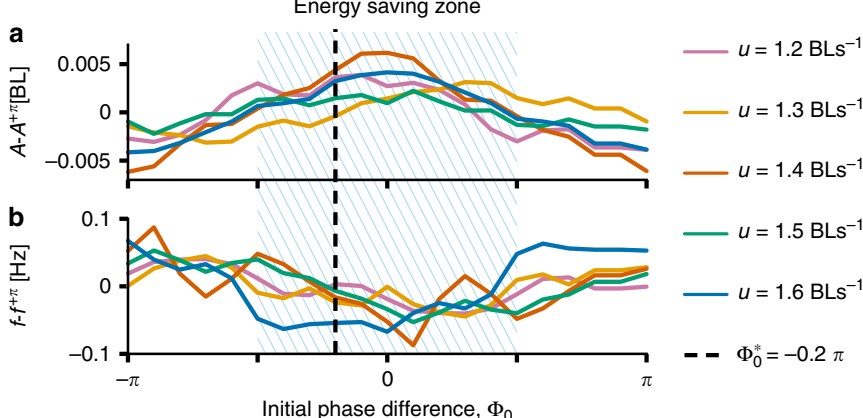

**Fig. 4 Relative energetic benefits to a follower in real fish pairs. a, b** Energy cost analysis was conducted by calculating the difference in amplitude $A$ (**a**) and frequency $f$ (**b**) at $\Phi_0$ and the same measurements with the opposite phase $\Phi_0 + \pi$ (written as $A^{+\pi}$ and $f^{+\pi}$ respectively) as a function of initial phase difference $\Phi_0$ (Supplementary Fig. 25 and Note 4). Data are pooled from all pairs when the follower's front-back positions are not >0.4 BL distance (where the hydrodynamic interactions are expected to be the strongest). The hatched areas show the energy saving zone of $\Phi_0$. The dashed line denotes $\Phi_0^*$, the most typically observed initial phase difference exhibited by our fish. (Average amplitude is 0.09 BL, average frequency is 2.3 Hz).

interactions are mainly determined by the initial phase difference $\Phi_0$ (see above), we analysed performance in the full possible range from $-\pi$ to $\pi$ (see Supplementary Fig. 25 for the detailed method). We define relative energy saving when fish exhibit higher tailbeat amplitudes $A$ (Fig. 4a) and lower tailbeat frequencies $f$ (Fig. 4b) than average[48], and find that the range is $\Phi_0 \in [-0.5\pi, 0.5\pi]$ (the shaded area in Fig. 4). Figure 4 also shows that while fish adopting $\Phi_0 \approx 0$ will save the most energy, those exhibiting $\Phi_0 = -0.2\pi$, as in our experiments, will save almost the same amount (thus they are very close to optimal in this respect).

Fish in our experiments (Fig. 3c at $D = 0$ BL) spent 59% of their time swimming with phase relationships ($\Phi_0 \in [-0.5\pi, 0.5\pi]$) that save energy, and the remaining 41% that imposes some (relative) energetic costs. However, because the energy cost has a sinusoidal relationship to the phase difference (Fig. 1d) simply calculating the percentage of time in each regime (in which there is either a benefit or a cost, regardless of the magnitude of each) is insufficient. By combining the frequencies (occurrences) of each phase difference $\Phi$ observed in Fig. 3c and the sinusoidal shape of the power cost as a function of $\Phi$ (Fig. 1d and Supplementary Fig. 6b, d), we can estimate that by behaving as they do, fish (in our flow conditions) save (by accumulating all benefits and extra costs; where a random behaviour would give 0) an overall 15% of the total possible (which would be achieved by perfectly adopting the optimal phase to the neighbour-generated vortices at all the time, Supplementary Fig. 27). It is possible that if fish are exposed to more challenging, stronger flow regimes (here we employed those of typical swimming), that this percentage will increase. However we would never expect fish motion to be completely dominated by a need to save energy as they must also move in ways as to obtain salient social and asocial information from their visual, olfactory, acoustic and hydrodynamic environment, such as to better detect food[52], environmental gradients[44] and threats[16]. Nevertheless, kinematic analysis suggests that they adopt VPM in a way that results in energy savings (dashed line in Fig. 4).

In summary, our bio-mimetic robots provided an effective platform with which we could explore the energetic consequences of swimming together in pairs and revealed that followers could benefit from neighbour-generated flows if they adjust their relative tailbeat phase difference linearly as a function of front-back distance, a strategy we term vortex phase matching. A model based

on fundamental hydrodynamic principles, informed by our flow visualisations, was able to account for our results. Together, this suggests that the observed energetic benefit occurs when a follower's tail movement coincides with the induced flow generated by the leader. Finally, experiments with real fish demonstrated that followers indeed employ vortex phase matching and kinematic analysis of their body undulations suggests that they do so, at least in part, to save energy. By providing evidence that fish do exploit hydrodynamic interactions, we gain an understanding of important costs and benefits (and thus the selection pressures) that impact social behaviour. In addition, our findings provide a simple, and robust, strategy that can enhance the collective swimming efficiency of fish-like underwater vehicles.

## Methods

**Experiments with robotic fish.** Experiments with robotic fish were conducted in a low turbulence flow tank at the College of Engineering, Peking University. The experimental apparatus is around 2-m-wide, 20-m-long and 10-m-high, and test area is ~0.4-m-wide, 1-m-long and 0.45-m-deep[53]. Our robotic fish were placed in the flow tank and suspended from a six-axis control platform (Supplementary Fig. 3 and Movie 2). This system allowed us to control each robot's position in three dimensions, alter the phase and frequency of the tail undulations, and measure the power consumptions (Supplementary Fig. 4). The phase difference $\Phi$, was controlled by initialising the movement of the follower robot after a specific time delay from the leader robot. For example, according to the definition of the body phase (Fig. 1c), a phase difference of $\pi$ was generated by adding a half period of delay to the follower robot. The robot was powered at 6 V by a stabilised power supply (RIGOL DP832). We sampled the current required to power the fish at 5000 Hz using a current acquisition device from National Instruments (NI 9227). To minimise noise and error from initialisation effects, data were collected only after both 10 s had passed, and at least five full undulations had occurred, and we recorded the average power cost (current × voltage) across 10 subsequent undulations. Each test was repeated five times and the efficiency coefficient was evaluated with average power cost of five repetitions. Experiments were conducted during 20 November 2014–05 December 2014.

During the experiments, the flow speed of the flow tank was set to the swimming speed of the robot in static water (0.245 ms$^{-1}$). In order to measure and remove baseline servomotor power costs, we conducted experiments both in air (suspended above the flow tank) and water (20 cm below the surface, 20 cm above the bottom and 15 cm away from the side boundary). In total, we conducted 10,080 experiments with two robots swimming alone in air, alone in water and schooling in water with various phase differences ranging from 0 to $2\pi$ with an interval of $0.2\pi$, front-back distance $D$ ranging from 0.22 to 1 BL with an interval of 0.022 BL and left-right distance $G$ ranging from 0.27 to 0.33 BL with an interval of 0.022 BL.

**Experiments with real fish.** Experiments with goldfish were conducted in a flow tank (Loligo system, Tjele, Denmark) at the Max Planck Institute of Animal Behavior in Konstanz, Germany. The effective test area of the flow tank is 0.25-m-wide, 0.875-m-long and 0.25-m-deep. A mirror was put at the bottom of the tank

at 45° respect to the horizontal plane to allow the camera to record the bottom-view from one side of the tank (Supplementary Fig. 15). Bottom-view and lateral-view cameras (BASLER acA2000-165umNIR) were used to film the fish movements at 100 frames per second. The resolution was set as 2048 × 1058. Videos were collected using a commercial code (Loopbio, Austria).

Before the experiments, we calibrated the flow speed with a vane wheel flow probe (Hontzsch, Germany). For each experiment, we randomly picked two fish, inhibited the lateral line if necessary and moved them to an acclimation container, which was slowly filled with water from the flow tank. After 30 min, and if the temperature difference between the flow tank and the acclimation container was <2 °C, we transferred both fish to the flow tank. We left the fish inside the flow tank for another 30 min without flow. During this period, several snapshots were taken to measure body length of each fish. Flow speed was set according to the average body length of the fish and ranged from 1.2 BLs$^{-1}$ to 1.6 BLs$^{-1}$ with an interval of 0.1 BLs$^{-1}$. White light was given for the intact (V+LL+) and lateral line impaired fish (V+LL-), and infrared light (850 nm) was given for the vision impaired intact (V-LL+) or both vision and lateral line impaired fish (V-LL-; the methodology for lateral line impairment is detailed in the Supplementary Note 4). We recorded the swimming of fish pairs in the flow tank for 5 min with different flow speeds (in a randomised order) with 5 min resting periods without flow in between.

**Data analysis**. We first detected the positions of each fish in each frame of all the videos (both bottom and side views) with DeepLabCut[54]. Then precise tracking was achieved by Kalman filtering and applying a greedy algorithm (a local optimisation method) to the detected positions. We then cropped fish images according to the tracking points and detected each fish posture with DeepPoseKit[47]. Finally, we applied a Hilbert transformation[55] to obtain body phase $\phi$ and amplitude $A$, and wavelet analysis to obtain frequency $f$ (Supplementary Fig. 16). The centre of the fish body was estimated as being one third of the fish body length from the tip of the head. Left-right distance $G$ was calculated as the distance between the centre points in the $y$-axis (lateral to the flow direction). Front-back distance $D$ was calculated as the difference between the centre points in the $x$-axis (in the direction of the flow).

Based on the tracking results, we filtered the data to extract paired swimming, where the relative position of the follower could allow a possible interaction with the vortex shed by the leader. From 52 h of videos, we collected 133.5 minutes of paired swimming in total, 287,818 frames for the intact group (V+LL+), 142,645 frames for V-LL+, 170,870 frames for V+LL- and 199,383 frames for V-LL-. For each case, we split the data according to the front-back distance $D$ with 20 bins. Within each bin we computed the density distribution $P(\Phi|D)$ histograms from which the resulting heatmaps (over the range of front-back distance $D$), were generated.

The theoretical phase difference was calculated according to Eq. (2) (in the main text) with $\Phi_0 = \Phi_0^*$. $\Phi_0^*$ is determined by following steps: First, we calculated for each front-back distance the phase difference $\Phi^*$ that fish adopted with highest probability using circular statistics and Von Mises fitting[56]; then we used a periodic least square fitting to obtain $\Phi_0^*$.

Randomised data were generated by swapping one fish's postures between two randomly selected trials within the respective treatment (Supplementary Fig. 20), following which we filtered data following the same criteria in the Supplementary Note 4. We conducted 1000 randomisations for all cases (intact and three impaired treatments). Heatmaps were generated with the same method described above.

All animal handling and experimental procedures were approved by Regierungspräsidium Freiburg, 35-9185.81/G-17/90.

**Reporting summary**. Further information on research design is available in the Nature Research Reporting Summary linked to this article.

## Data availability
The data that support the findings of this study are available in figshare with the identifier https://doi.org/10.6084/m9.figshare.12762107 1 Source data are provided with this paper.

## Code availability
All the data analyses were performed using custom scripts written in MATLAB (MathWorks Inc., Natick, MA, USA) and Python (Python Software Foundation, 2018). All codes that support the findings of this study are available in figshare with the identifier https://doi.org/10.6084/m9.figshare.12762107 Source data are provided with this paper.

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

## Acknowledgements

We thank members of the Department of Collective Behaviour and the Intelligent Biomimetic Design Lab who assisted with the project, especially L. Jia for his constructive discussion of the hydrodynamic analysis, A. Liu, B. Fu and R. Mao for their help of conducting robotic fish experiments, and R. Tian and L. Zhang for their support of the CFD codes and help of simulations. L.L. acknowledges J. Zhang, X. Zhang, M.S. Triantafyllou, D. Weihs, E. Kanso, J. Davidson, G. Amichay, R. Ravi and A. Jordan for their constructive discussions on this project, and C. Chen for her encouragement. I.D.C. and L.L. gratefully acknowledge fish care and technical support from M. Mende, M. Miller, D. Piechowski, M. H. Ruiz, C. Bauer, J. Weglarski, A. Bruttel and D. Leo. We acknowledge funding from the Max Planck Institute of Animal Behavior. G.X. acknowledges support from the National Natural Science Foundation of China (NSFC, No.61973007, 61633002) and the Beijing Natural Science Foundation (No.4192026). I.D.C. acknowledges support from the NSF (IOS-1355061), the Office of Naval Research grant (ONR, N00014-19-1-2556), the Struktur-und Innovationsfunds für die Forschung of the State of Baden-Württemberg, the Deutsche Forschungsgemeinschaft (DFG, German Research Foundation) under Germany's Excellence Strategy–EXC 2117-422037984, and the Max Planck Society. M.N. acknowledges funding awarded by the Hungarian Academy of Sciences (a grant to the MTA-ELTE 'Lendület' Collective Behaviour Research Group). L.L. acknowledges the support from Zukunftskolleg Independent Research Grant (P82967018 FP 670/18) and the support of NVIDIA corporation with the donation of a Titan Xp GPU.

## Author contributions

L.L., G.X. and I.D.C. conceived the idea and designed the project; L.L. conducted the experiments and collected the data; L.L., M.N. and G.X. analysed the robotic fish data and derived the rule; L.L., M.N., J.M.G., J.B. and I.D.C. analysed the real fish data; L.L., M.N., J.M.G., J.B. and I.D.C wrote the initial draft of the manuscript and all authors revised the manuscript.

## Funding

## Competing interests

The authors declare no competing interests.
