## [Peer Review File · Nature Communications]

Reviewers' comments:

Reviewer #1 (Remarks to the Author):

Dear Editor and Authors:

Thank you for the opportunity to review the paper "Schooling fish save energy by vortex phase matching" by Li et al., which describes experiments on group swimming of robotic and actual fish. Focusing on pairs of swimmers, the work presents power measurements on a follower robot behind a leader to show that specific phase relationships between flapping motions and oncoming vortices can save on the energetic costs of swimming. Further, experimental measurements on the motions of swimming fish are interpreted as showing a preference for such "vortex phase matching".

The work aims to address an important and long-standing question that touches on many fields of science and engineering: Do fish take advantage of interactions through flows during schooling? I am big fan of the main motivations and objectives of the work. And the authors should be commended for their bravery in wading into what has long been an extremely controversial and challenging problem. And they do so with a refreshing combination of physical and biological experiments – such an interdisciplinary approach is powerful and useful but not at all easy for any one team of researchers to pull off. And I especially like that the authors sought out a relatively simple phenomenon within this complex problem, namely that of phase matching between swimming motions and ambient vortices.

This being said, I think the presentation in the manuscript has much room for improvement. I feel that many important aspects of the data and data analysis are buried in the supplement. Portions of the paper are left so under-explained and under-substantiated that I found myself questioning the main conclusions drawn by the authors. Also to this point, I feel like some of the main quantitative results are never directly stated and interpreted, and these are important for the reader to assess the significance of the findings. I detail my major criticisms below, and also I point out some more minor comments and suggestions. Overall, I feel the work is not ready for publication as it stands, but I would happily review a revised version with significant improvements. The paper could be impactful within the field, but this hinges very much on a clear, direct and honest presentation of the findings.

Major comments:

1) My main criticism about the presentation is that the main paper does not stand on its own. Many key steps in drawing the central conclusions are buried in the supplement, which itself is not so easy to read. My recommendation is that the authors move to the main paper all data and analyses that are essential to understanding the main conclusions and essential for evaluating the significance of these conclusions. For example, the main result of the robotic experiments seems to be Fig. 2A. But we do not see any of the underlying data, such as swimming kinematics and power measurements, that go into this plot. This makes it very difficult to assess the results. For example (elaborated below), are the few percent energy savings of Fig. 2A at all significant when compared with the errors in the power measurements? It is difficult to tell without ever seeing the power traces. In general, the reader deserves direct access within the main paper to all basic data and to the intermediate steps in the data analysis leading to the final concluding plot. I think this should apply to both sets of experiments, and the figures should logically and systematically take the reader through the raw measurements and the analyses.

2) The order of presentation for the experiments also seems a little backwards to me. Specifically, there seem to be two main results: 1) schooling fish lock flapping phase with oncoming vortices and 2) robotic measurements show such vortex phase matching can save energy. I think this order will make more sense to the reader, since the robotic experiments are really intended as an

explanation for the biological observation. (Perhaps the current order reflects the actual chronology of the work, but this does not seem so important as conveying the message clearly.) My feeling is that the paper will be easier to present in this new order and certainly easier to read.

3) The vortex phase matching phenomenon is particularly difficult to decipher from the authors' presentation. It seems there is a far simpler way to explain it that is closely based on previous works, some of which have gone uncited in the paper. The key missing concept is spatial (as opposed to temporal) phase. Each swimmer's tail traces a sinusoidal path through the fluid, leaving a similarly spatially periodic wake of vortices. A follower's oscillatory path then assumes some spatial phase difference or spatial shift relative to a leader's path, and this phase difference dictates the follower-wake interactions. I think what the authors call " ϕ_0 " is something like this spatial phase. It nicely combines the effects of temporal phasing from flapping and the phase shift induced by the time elapsed between the leader shedding a vortex (or any flow structure) and the follower encountering it. If my understanding is correct, the paper and its results could be greatly simplified by introducing this concept. It would also relate to several earlier papers on closely related problems: Portugal Nature 2014, Zhang PRL 2015, Becker Nat. Comm. 2015, Gravish PRL 2015, Ramanarivo PRF 2016, Newbolt PNAS 2019 and others. The papers by Portugal, Gravish and Newbolt in particular explain this quantity and its significance, and I think these and any related works should be properly cited when the VPM model is introduced. This would save a lot of text in the mechanism section, and corresponding changes in Figs. 2 and 3 would make these data much easier to interpret.

4) Some of the most basic quantitative results are never clearly stated, and these are very important for readers in assessing the significance of the findings. For one, it seems from Fig. 2A that the power savings available are on the order of a few percent of the power expended during lone swimming. This basic number should be stated clearly in the text when reporting the results as well as in the discussion. It is so fundamental that it belongs in the Abstract as well. Whether a few percent is interpreted as "small" or "significant" may depend on the reader – for the airline industry, a 1% fuel savings would represent a huge breakthrough! – but in any case the fundamental numbers should be given clearly and directly. Secondly, it seems from Fig. 3 that fish occupy the energetically favorable positions only a few percent more often than other positions. (The presentation of these results is quite unclear and indirect, however, so my understanding may be off.) This too should be clearly stated in the results section, in the discussion section and also in the Abstract. The text as written – e.g. from the Abstract, "... fish adopt this same vortex phase matching relationship..." – could be viewed as quite misleading.

Considering the result that fish occupy favorable configurations with a few percent preference, together with the result that these configurations represent a few percent savings, one can estimate the savings associated with schooling as multiplicative in these quantities and thus very small, well under 1%. I think the authors should openly and honestly discuss these results and their interpretation. Would the authors conclude that the hydrodynamic benefits are quite minimal? And why then do fish seem to show a preference for certain interactions with flows (i.e. Fig. 3)?

5) It is not at all clear if the actual fish display phase locking that is quantitatively similar to the phase locking that is found to be energetically favorable in the robotic fish system. Here the presentation is vague and should be made direct and clear, and the result should be more completely interpreted. In particular, it seems from Fig. 2A that the spatial phase $\phi_0 = 0.3\pi$ is optimal and yet from Fig. 3 that fish adopt $\phi_0 = -0.2\pi$. The difference of about 0.5π seems significant given that a change of ϕ by π changes the savings from most optimal to most detrimental (energy losses). In any case, I think an open, honest and complete discussion would help the reader make sense of the results. Do actual fish adopt the optimal spatial phase? If not, what are the savings found in the robotic system for the spatial phase that the actual fish do adopt? Can any differences in results be explained by differences in the two systems? How do the authors interpret these findings for schooling more generally?

Some more minor comments:

- 1) Title: Have the authors really shown this? It seems like an overstatement given that no energy measurements on fish are performed. I think a more honest title conveying more directly what is actually shown in the work would be more useful and still be impactful.
- 2) Abstract: In addition to the points given in the Major comments above, it seems the focus on the two-fish situation should be stated. Also, there does not seem to be any broader interpretation of the results for fish schools.
- 3) Introduction: The background review is quite detailed, but one common thread in many recent papers seems to be left out. Namely, many papers from the last ~5 years have shown that specific phase relationships can come about entirely passively from the flow interactions: Zhang PRL 2015, Ramanarivo PRF 2017, Newbolt PNAS 2019 and others. This represents an alternative to the view that specific phases or positions must be achieved through active sensing-and-response behaviors. In fact, it seems that the authors' observations on actual fish might be well explained by this hypothesis, especially given that fish display vortex locking even without vision and without the lateral line. In general throughout the paper, the authors should consider more discussion along these lines. For example, if fish adopt a different phase from what is optimal, then could it be that they adopt a phase that is passively preferred due to flow interactions?
- 4) The very recent work by Oza PRX 2019 also seems highly relevant, especially in regard to different schooling configurations.
- 5) Page 6 first sentence of Results section: "costs" might be replaced with "benefits" or, more neutrally, "costs and benefits"
- 6) Page 6 near bottom: "swimming speed/flow speed" should be "swimming or flow speed"
- 7) The fact that the robots are fixed in a flow and not free swimming should be made clear. Given this, it seems that the robots are not in mechanical equilibrium in general, and so the conditions explored in the experiments would not be realized by free swimmers.
- 8) The quantity called efficiency is not a true efficiency but more like a dimensionless power consumption, normalized by that of a single swimmer. The authors may consider a different name.
- 9) Sentences spanning pages 7 and 8 on range of interaction: This seems like an overstatement given that the breakdown of a wake depends sensitively on aspects such as flapping kinematics and tail geometry. I would suggest refraining from making general claims based on one set of such parameters.
- 10) Page 8, reference to figures 7 and 8: These are not in the main paper, so perhaps a citation to the supplement is intended here?
- 11) Fig. 1 seems to emphasize flow visualization, which is not at all used in anyway throughout the rest of the work. Missing is any information about the actual kinematics, space of configurations explored, sample power traces, etc. These are much more important data for the plots to come.
- 12) Fig. 2C and caption: As in other parts of the paper, it seems spatial phase could tighten up the explanation here. It seems the authors are saying something simple about spatial phase, but I was not able to grasp the point of the schematics and very long caption.
- 13) Fig. 3: The plots (together with vague text) are a little misleading, given the colorbar runs over just a few percent in terms of occupancy probability (if this is in fact what is meant by the quantity P).
- 14) The plots of Figs. 2 and 3 should be brought into alignment in terms of axes range and such. For example, D should go from 0 to 1 in these plots (leaving blank $D < 0.22$ if this was not explored in Fig. 2) and ϕ from 0 to 2π . The repetition of the same data in Fig. 3 for ϕ outside this range seems strange and not useful. Uniformity of the plots is important to, for example, compare the savings bands of Fig. 2 with the occupancy bands of Fig. 3.
- 15) I found the term "socially-generated flow" to be odd since the source is physical and not social or behavioral – "collectively generated"?
- 16) Page 13, reference to reflex or proprioceptive response – couldn't all of these data also be explained by the simple hypothesis that phase locking comes about passively/physically through flow interactions? See minor point 3 above.
- 17) I didn't understand the reasoning regarding Fig. 4. These data seem to have no obvious connection to energy savings, yet it is claimed (in the caption and text) that "fish save energy". I

think this portion of the paper needs to be carefully reworked in terms of what is trying to be conveyed. The direct message seems to be that fish have different flapping frequency and amplitude when in different spatial phase relationships. But what more can be said? Also, these changes measured are very slight – do they matter? Again, more discussion about quantitative results and their interpretation is needed.

18) First sentence of Conclusions: “costs” might be replaced with “benefits” or, more neutrally, “costs and benefits”

19) Methods section: This does not seem to be about methods but reads like a summary of the paper.

Reviewer #2 (Remarks to the Author):

Referee Liang li and others, schooling Fish save energy by vortex phase matching

This is an interesting paper in which energy loss is measured for pairs of robotic fish. Results are used to develop a mathematical model of matching the phase of vortices in order to save energy when swimming in a pair.

Predictions of this model are subsequently tested in pairs of real fish and confirmed.

This method of experiments with robotic fish, mathematical model and empirical testing is very good. The paper needs improved in that it should state more clearly already in the abstract, and also throughout the main paper, that results merely concern pairs of fish not schools.

The main paper should also be more clear regarding:

1. What is meant with constructive and destructive reaction to vortices by the follower, see for instance, caption of figure 2 C-E.
2. As regards details of phase matching of vortices, in the last paragraph of page 7, it is mentioned that the benefits of swimming together decay fast with increasing distance of follower to leader, figure 2A. It needs more explanation how we can observe this in fig 2A. It should be explained what the efficiency coefficient refers to and what positive and negative values imply. Also what is meant in fig 2 CDE should be explained in more detail.
3. There is no mention of the size of the tank relative to the robotic fish and real fish, but this may have a large effect on flow. For instance first paragraph on page 7 of main text should inform on this and on the effects of walls on flow.
4. The relation between energy saving zone in figure 4 and vicinity of vortices in text second paragraph below figure 4 should be made clear.

The manuscript needs also improved as to its references, sometimes pairs are cited that have no relation to the text, for instance, in introduction page 3, in the second paragraph reference 14 is incorrectly located, e.g., page 4 last paragraph Boschitsch et al is mentioned with an nonmatching reference to 23.

The authors need to go carefully through the complete text to amend references.

Reviewer #3 (Remarks to the Author):

This is the review of NCOMMS-19-363561 titled “Schooling fish save energy by vortex phase matching”. In this manuscript the authors investigate the energy saving mechanisms of fish schooling through experiments with robots and real fish. In the robotic fish experiments, a pair of fish are placed at different lateral and front-back space D with different phase angles while the energy cost is measured. It was found that the power was more sensitive to the front-back spacing and phase angle than the lateral spacing. The length of the fish, tail beat amplitude, and frequency were kept constant. It was found that the phase difference proportional to the fD/u minimized

energy consumption. It was suggested that the mechanism for energy reduction is constructive vortex interaction. This idea was also tested on a small number of goldfish pairs. The phase of the goldfish pairs and their distance D , qualitatively agreed the robotic experiments (not as good though).

I do not recommend this paper for publication because of the following reasons:

- 1) The interaction is only investigated in pairs rather than in a school. So the title can be misleading.
- 2) I have a few concern regarding the vortex phase matching hypothesis: a) it does not consider the lateral movement of the vortex. In fact, for constructive interaction, the location of the similar-sign vortices should be similar both in axial and lateral locations; b) It is only tested for one velocity and frequency (u and f), i.e., out of the three parameters fD/u only one is varied. The dependence at other velocities and frequencies is not shown; and c) the speed of vortices is taken to be similar to the swimming speed. This has not been justified. Because of these, I do not feel the vortex phase matching is adequately tested.
- 3) It is not clear if the constructive vortex interaction occurs at all time instants during a tail beat or just at specific instants. If it happens at all time instants, it can also increase the drag (in addition to thrust). There are two force peaks per tail beat corresponding to back and forth motion of the tail in each tail beat. If constructive interaction occurs, it can increase maximums as well as the minimums during a cycle.
- 4) My main concern regarding the results of the robotic experiments is that the robotic fish are not self-propelled. In fact, it is not clear if the average force over the fish are zero over the cycle. The energy saving during free swimming and tethered conditions might not be the same.
- 5) The mechanism described does not clearly show the constructive interaction of vortices. In fact, the main figure 2 and 5 (supplementary) does not show the vortices of the follower fish. This makes me wonder if the mechanism observed by the authors is due to vortex interaction. As an example, I can provide another hypothesis on the energy saving based on the same figure: the main portion that generate thrust in such fish is the tail. The follower is positioned such that the induced flow in the wake of the leader increases the angle of attack to the tail of the follower and, consequently, increase the generated thrust.
- 6) The flow visualization of interaction of the fish using hydrogen bubbles show instabilities in the flow which does not seem realistic for actual fish probably because of the tethered nature of the fish. It is very different than the reverse von karman for a single fish.
- 7) I am not convinced by the experiments with real fish because as mentioned in the previous comment, the mechanism might be something other than vortex interaction. In fact, the relation might be true but for other reasons than vortex interaction.

Reviewer #1 (Remarks to the Author):

R1.Q1:

Dear Editor and Authors:

Thank you for the opportunity to review the paper “Schooling fish save energy by vortex phase matching” by Li et al., which describes experiments on group swimming of robotic and actual fish. Focusing on pairs of swimmers, the work presents power measurements on a follower robot behind a leader to show that specific phase relationships between flapping motions and oncoming vortices can save on the energetic costs of swimming. Further, experimental measurements on the motions of swimming fish are interpreted as showing a preference for such “vortex phase matching”.

The work aims to address an important and long-standing question that touches on many fields of science and engineering: Do fish take advantage of interactions through flows during schooling? I am big fan of the main motivations and objectives of the work. And the authors should be commended for their bravery in wading into what has long been an extremely controversial and challenging problem. And they do so with a refreshing combination of physical and biological experiments – such an interdisciplinary approach is powerful and useful but not at all easy for any one team of researchers to pull off. And I especially like that the authors sought out a relatively simple phenomenon within this complex problem, namely that of phase matching between swimming motions and ambient vortices.

This being said, I think the presentation in the manuscript has much room for improvement. I feel that many important aspects of the data and data analysis are buried in the supplement. Portions of the paper are left so under-explained and under-substantiated that I found myself questioning the main conclusions drawn by the authors. Also to this point, I feel like some of the *main quantitative results are never directly stated and interpreted*, and these are important for the reader to assess the significance of the findings. I detail my major criticisms below, and also I point out some more minor comments and suggestions. Overall, I feel the work is not ready for publication as it stands, but I would happily review a revised version with significant improvements. The paper could be impactful within the field, but

this hinges very much on a clear, direct and honest presentation of the findings.

R1.A1:

We thank the Reviewer for the positive and constructive comments and many helpful suggestions. We have rewritten the paper according to these comments.

Major comments:

R1.Q2:

1) My main criticism about the presentation is that the main paper does not stand on its own. Many key steps in drawing the central conclusions are buried in the supplement, which itself is not so easy to read. My recommendation is that the authors move to the main paper all data and analyses that are essential to understanding the main conclusions and essential for evaluating the significance of these conclusions.

R1.A2:

We agree. As the Reviewer noted we have adopted a highly multidisciplinary approach to tackle this long-lasting problem. Consequently we are presenting a much larger body of work than is typical for a single paper and we were aiming to make our work as easy to read for a diverse readership. We tried to keep the manuscript as brief and concise as possible, but in doing so we inadvertently put some interesting and important parts of our work into the SI. We agree with the Reviewer that we went too far in cutting out parts from the main text and showing them only in the SI such as the information regarding the steps of data collection and analyses. We have conducted a major revision of the article (blue font) including substantive changes in the main text and figures, and we added new analyses where it was requested. We have moved several parts, including all essential data and analyses, from the SI into the main body of the paper to aid clarity.

For example:

1. *We report exact values in the text; for example in the abstract (Line 28: “Our results demonstrate that robotic followers could obtain energetic benefits (up to 13% relative to swimming alone) from a near neighbour...”) and results sections (Line 248: “The difference between the maximum energy saving and maximum energy cost for the robots is around 13.4%.”).*

2. *Main Fig. 1 is updated with a detailed presentation of the measured variables and an example of the experimental test with the robotic fish. We included the measured power costs as a function of the phase difference.*
3. *We add more, and clearer, descriptions regarding the real fish data analyses. For example:
Line 375: "... For each front-back distance, we show the distribution of the phase difference of the fish (each column in Fig. 3b and c). The peak of the distribution of the phase difference is linearly correlated to the front-back distance (the bright area, Fig. 3b and c). To test whether the observed linear correlation corresponds with vortex phase matching, we predicted the phase difference that, if employed by the fish, would minimise energetic costs based on Eq. 2 for each front-back distance with same $\phi_0 = \Phi_0^* = -0.2 \pi$ as in the data (Fig. 3d). ..."*
4. *A discussion of the relative ratio of the energy saving in SI has been moved to the main text.*

R1.Q3:

For example, the main result of the robotic experiments seems to be Fig. 2A. But we do not see any of the underlying data, such as swimming kinematics and power measurements, that go into this plot. This makes it very difficult to assess the results.

R1.A3:

Thank you for making this point. A description of the kinematic analyses has now been added in the main text (Line 176: "... the kinematics that mimic normal real fish body undulations when swimming³⁵ (see Supplementary Fig. 2 for a detailed comparison of the kinematics; parameters of the controller are given in the Supplementary Information).") and a comparison of body kinematics is added in Supplementary Fig. 2. The power cost of the robotic fish as a function of the phase difference is now given in the main Fig. 1d. This is a new plot showing an energy saving for a front-back distance D . This also intends to convey a clear message to the reader about the magnitude of energy savings possible (as raised in later points, below).

R1.Q4:

For example (elaborated below), are the few percent energy savings of Fig. 2A at all significant when compared with the errors in the power measurements? It is difficult to tell without ever seeing the power traces.

R1.A4:

We added new figures in the main text and supplementary information (Fig. 1d and Supplementary Fig. 8). It is shown that we replicated the measurement 5 independent times for each phase difference. The variation (caused by noise and errors in the measurement) for each phase difference is depicted by the shaded area, which indicates the noise of the system. The phase difference has a much larger effect on the power cost of the follower robot than does noise/error.

R1.Q5:

In general, the reader deserves direct access within the main paper to all basic data and to the intermediate steps in the data analysis leading to the final concluding plot. I think this should apply to both sets of experiments, and the figures should logically and systematically take the reader through the raw measurements and the analyses.

R1.A5:

We agree. As described above we have extensively revised the main text in light of the reviewer's comments to show the steps of the data analysis. We also moved detailed methods to the main text. We again thank the Reviewer as this has greatly improved clarity.

R1.Q6:

2) The order of presentation for the experiments also seems a little backwards to me. Specifically, there seem to be two main results: 1) schooling fish lock flapping phase with oncoming vortices and 2) robotic measurements show such vortex phase matching can save energy. I think this order will make more sense to the reader, since the robotic experiments are really intended as an explanation for the biological observation. (Perhaps the current order reflects the actual chronology of the work, but this does not seem so important as conveying the message clearly.) My feeling is that the paper will be easier to present in this new order and certainly easier to read.

R1.A6:

We thank the Reviewer for this suggestion. We also thought about this logic before, and we agree that paper structure should not necessarily follow the

chronology of the work undertaken. However, the main reason we chose current logic is that the experiments with robotic fish were both designed to, and employed to, **generate testable hypotheses** regarding what we may expect to see in the real biological system. One of the great challenges in conducting biological experiments is knowing what to look for. Without a clear hypothesis it is very difficult to make progress. This is perhaps one of the reasons that such little progress has been made in addressing the fundamental question: do fish save energy when swimming together? Biologists haven't necessarily known what to look for in their experimental data. The robotic experiments helped us reveal a biologically-plausible (and simple) rule that if followed by real fish, would allow them to save energy. It also revealed to us how we would expect factors like turbulence to result to a dramatic decay of possible benefits as a function of front-back distance, and thus guided us to focus on vortex-body interactions (as opposed to second order vortex-vortex interactions).

We think that changing this logic would introduce some weaknesses into the flow of the paper. Specifically, no robot fish, no matter how well constructed, can fully approximate the sophistication of real fish (which have evolved exquisite adaptations to their environment evolved over millions of years that we can only approximate with even the most sophisticated robotics). We feel that if we reversed the order of our paper the reader may be misled to think that the robotic experiment was intended to mimic **exactly** the biological system. The similarity of the results from robotic and real fish data provides evidence that the VPM is both very general and very robust (our robot was designed to follow typical swimming patterns exhibited by real fish (Line 176: "... the kinematics that mimic normal real fish body undulations when swimming³⁵ (see Supplementary Fig. 2 for a detailed comparison of the kinematics; parameters of the controller are given in the Supplementary Information).") -- and that the behavioural rule a biological system would need to employ is simple, and thus may be generic among many species of schooling fish (we also note here that "schooling fish", as employed in the biology literature, refers to species of fish that can form schools, not that they necessarily are in a school at a given time). There are, of course, some deviations between the two systems due to the differences between them. In addition, if we would implement the new logic, the comparison between biological data and theoretical prediction will be split into two parts at the beginning and end of the paper, and we feel this will make the paper less cohesive.

R1.Q7:

3) The vortex phase matching phenomenon is particularly difficult to decipher from the authors' presentation. It seems there is a far simpler way to explain it that is closely based on previous works, some of which have gone uncited in the paper. The key missing concept is spatial (as opposed to temporal) phase. Each swimmer's tail traces a sinusoidal path through the fluid, leaving a similarly spatially periodic wake of vortices. A follower's oscillatory path then assumes some spatial phase difference or spatial shift relative to a leader's path, and this phase difference dictates the follower-wake interactions. I think what the authors call "phi_0" is something like this spatial phase. It nicely combines the effects of temporal phasing from flapping and the phase shift induced by the time elapsed between the leader shedding a vortex (or any flow structure) and the follower encountering it. If my understanding is correct, the paper and its results could be greatly simplified by introducing this concept. It would also relate to several earlier papers on closely related problems: Portugal Nature 2014, Zhang PRL 2015, Becker Nat. Comm. 2015, Gravish PRL 2015, Ramanarivo PRF 2016, Newbolt PNAS 2019 and others. The papers by Portugal, Gravish and Newbolt in particular explain this quantity and its significance, and I think these and any related works should be properly cited when the VPM model is introduced. This would save a lot of text in the mechanism section, and corresponding changes in Figs. 2 and 3 would make these data much easier to interpret.

R1.A7:

We agree. This is a terrific suggestion and we thank you for directing our attention to the missing citations. There is a clear relation between these and this point is now made in the new version. We agree with the Reviewer, that our description using "Phi_0" is similar to the concept of "spatial phase", as it describes the phase difference between the wake generated by the leader and the follower's body in space, and that making this explicit improves the clarity of the behaviour observed.

We revised the main text as:

Line 319: "The definition of the phase difference Φ in the VPM rule (Eq. 2) is analogous to the concept of "spatial phase", with the first term corresponding to the "spatial phase" contribution (how much the phase is changed as a result of the travel of the vortex) and the second term (initial phase difference Φ_0) corresponding to the "schooling number" SS

(with $S \bmod 1 = \Phi_0 / (2\pi)$)²⁹ (more details on this correspondence can be found in the Supplementary Information). We decided to use Φ_0 as it fits better the description of the biological system where each swimming characteristic is dynamically changing in time (such as the front-back distance D , undulation frequency f , etc.).”

And in the Supplementary Information:

Line 1276: “In our paper we used “initial phase difference” Φ_0 to describe the phase difference between the body and vortex shed by the leader (See Fig. 2c-e). “Spatial phase” and “ Φ_0 ” capture a similar concept, but for the “Lagrangian” and the “Eulerian” specifications of the flow field, respectively. We can describe the vortex-body interactions in either way, as both describe the same phenomena just using different coordinate systems.

If we track the wake and the body continuously and describe the path “drawn” by the tailtip (a sinusoidal track), it is better to describe in “Lagrangian specification” with “spatial phase”. If we are interested in the phase at a specific distance, however, it may be more intuitive to describe our analysis in the “Eulerian specification” with “ Φ_0 ”, the phase difference at front-back distance 0. In our case, since the concept of “Eulerian specification” is also applied in the following real fish data analysis, we adopted the initial phase difference throughout the paper.”

This is really subject to what may make the text more clear to the reader. We could describe it in either way. However, we have found, when discussing this work with biology colleagues, that they find the concept of “spatial phase” conceptually more challenging than employing Φ_0 , which describes the temporal relationship. But since the former may be more intuitive to other readers (physicists in particular), we agree that it is important to explain that the two are interchangeable.

If the Editor/Reviewer disagree, and think that a shortened version using only spatial phase would be as accessible for a wide audience (especially including biologists), we would be willing to adopt this suggestion (the two are, as we now write, interchangeable framings of the same phenomenon). We are very open to follow advice here.

R1.Q8:

4) Some of the most basic quantitative results are never clearly stated, and these are very important for readers in assessing the significance of the findings.

R1.A8:

We agree, and apologise for this oversight. Please see our response to major comment 1 (R1.A2). We have now added the basic quantitative results to the main text (e.g. in the abstract and result section, we reported the difference between maximum energy saving and maximum extra energy cost relative to swimming alone (Line 28: “Our results demonstrate that robotic followers could obtain energetic benefits (up to 13% relative to swimming alone) from a near neighbour in any relative position by maintaining constructive interactions with the shed vortices (termed ‘vortex phase matching’, VPM), a result that can be accounted for with a minimal model of hydrodynamic interactions.” and Line 248: “The difference between the maximum energy saving and maximum energy cost for the robots is around 13.4%.”).

R1.Q9:

For one, it seems from Fig. 2A that the power savings available are on the order of a few percent of the power expended during lone swimming.

R1.A9:

The relevant consideration of energy saving and energy cost is that relative to the power cost of swimming alone (natural selection typically operates on relative fitness, and thus relative costs and benefits - e.g. Sloan-Wilson (2004) “What is wrong with absolute individual fitness?” Trends in Ecology and Evolution). We have now made this clear in the text (see Line 244: “Correspondingly, the coefficient η compares the energy cost of fish swimming in pairs to swimming alone.”). We have reported explicitly the definition of η and provided a description in the Results section starting from Line 235. The values are reported in the abstract (Line 28: “Our results demonstrate that robotic followers could obtain energetic benefits (up to 13% relative to swimming alone) from a near neighbour...”) and discussed in a paragraph from Line 473: (“In our robotic experiments we find that a follower can save energy in any relative spatial position with respect to a near neighbour if it adopts a specific relative tailbeat phase difference, and that this depends predominantly on the front-back distance (Supplementary Figs. 9 and 10). The maximum energy saving of the follower robot was found to be lower than many previous predictions from theoretical models^{3,52} or

numerical simulations^{28,31}. This might be related to turbulence in the flow at high Reynolds number, and/or due to the effects of viscosity and the three dimensional nature of the flow in real systems. For example, Verma et. al¹⁰ found that the power costs of the follower can decrease up to 36% in 2D but only 5% in 3D CFD simulations.”)

R1.Q10:

This basic number should be stated clearly in the text when reporting the results as well as in the discussion. It is so fundamental that it belongs in the Abstract as well. Whether a few percent is interpreted as “small” or “significant” may depend on the reader – for the airline industry, a 1% fuel savings would represent a huge breakthrough! – but in any case the fundamental numbers should be given clearly and directly.

R1.A10:

We agree. In the new version, we have included all such basic number in the main text. For example, the percentage value of the difference between maximum energy saving and maximum extra power cost for robotic fish relative to swimming alone (13%) is reported in the abstract, results and discussions (R1.A2).

R1.Q11:

Secondly, it seems from Fig. 3 that fish occupy the energetically favorable positions only a few percent more often than other positions. (The presentation of these results is quite unclear and indirect, however, so my understanding may be off.) This too should be clearly stated in the results section, in the discussion section and also in the Abstract.

R1.A11:

Thanks for this comment. We agree that it is important to provide a more quantitative analysis and have done so in the revised version. Now, we have described this in the main text as-Line 443: (“Fish in our experiments spent 59% of their time swimming with phase relationships that save energy, and the remaining 41% that imposes some (relative) energetic costs.”) This indicates that fish spend 43% more ($59\%/41\%=1.43$) time doing energy saving than extra energy cost.

However, because the energy cost has a sinusoidal relationship to the phase difference (Fig. 1d) simply calculating the percentage of time in each regime (in which there is either a benefit or a cost, regardless of the magnitude of

each) is insufficient. To consider also the magnitude of the costs and benefits we have revised the main text as:

Line 449: “By combining the frequencies (occurrences) of each phase difference Φ observed in Fig. 3c and the sinusoidal shape of the power cost as a function of Φ (Fig. 1d and Supplementary Fig. 8b and d), we can estimate that by behaving as they do, fish (in our flow conditions) save (by accumulating all benefits and extra costs; where a random behaviour would give 0) an overall 15% of the total possible (which would be achieved by perfectly adopting the optimal phase to the neighbour-generated vortices at all the time; Supplementary Fig. 28).”

R1.Q12:

The text as written – e.g. from the Abstract, “... fish adopt this same vortex phase matching relationship...” – could be viewed as quite misleading.

R1.A12:

We have updated our text -this whole sentence now reads as:

Line 33: “Experiments with pairs of freely-swimming goldfish (*Carassius auratus*) reveal that, as predicted if employing VPM to save energy, fish typically adjust their relative tailbeat phase difference linearly as a function of front-back distance, allowing them to exploit neighbour-generated vortices.”

R1.Q13:

Considering the result that fish occupy favorable configurations with a few percent preference, together with the result that these configurations represent a few percent savings, one can estimate the savings associated with schooling as multiplicative in these quantities and thus very small, well under 1%.

R1.A13:

We apologise for the lack of clarity in our writing. As can be seen in main Fig. 4 there is a much broader range of phase difference values in which energy is still saved (denoted by the hatched area in that figure) and fish exhibit a (much) higher probability of being in this region (please see also R1.A11).

We agree with the Reviewer, we are also very interested and curious to measure directly how much (absolute) energy the fish can save due to

hydrodynamic interactions. However, in this study, our aim was to show convincingly that fish can and do save energy relative to swimming alone. But the quantitative question “exactly how much energy fish save” is out of the scope of the current study, as it is not possible for us (yet) to directly measure the power consumption of real fish. The best we can do is to give an estimation based on the real fish data (Fig. 3c) and the sinusoidal relationship between power cost and phase difference to estimate the percent (15%) of the total possible (Line 443: “Fish in our experiments (Fig. 3c at $D=0$) spent 59% of their time swimming with phase relationships ($\Phi_0 \in (-0.5\pi, 0.5\pi)$) that save energy, and the remaining 41% that imposes some (relative) energetic costs. However, because the energy cost has a sinusoidal relationship to the phase difference (Fig. 1d) simply calculating the percentage of time in each regime (in which there is either a benefit or a cost, regardless of the magnitude of each) is insufficient. By combining the frequencies (occurrences) of each phase difference Φ observed in Fig. 3c and the sinusoidal shape of the power cost as a function of Φ (Fig. 1d and Supplementary Fig. 8b, d), we can estimate that by behaving as they do, fish (in our flow conditions) save (by accumulating all benefits and extra costs; where a random behaviour would give 0) an overall 15% of the total possible (which would be achieved by perfectly adopting the optimal phase to the neighbour-generated vortices at all the time, Supplementary Fig. 28).”).

Were the fish to exhibit the same energy savings as did our robot (which is not possible as fish are far more efficient) it would be a savings of 1% (and not less) per tailbeat on average (a simple multiplication of 15% by $0.5 \times 13.4\%$). However, for the reasons outlined in our paper, and above, this will very likely be a considerable underestimate for the real biological system.

R1.Q14:

I think the authors should openly and honestly discuss these results and their interpretation. Would the authors conclude that the hydrodynamic benefits are quite minimal?

R1.A14:

We agree the hydrodynamic benefits for our robotic fish are smaller than one would expect based on previous 2D CFD studies (e.g. Hemelrijk et.al Fish Fish. 2014). However, a 3D CFD simulation also gives small amount of energy saving (see below).

Some potential reasons are mentioned above in (R1.A13) and paragraph in discussions (Line 473). In the new version, we discussed the difference between maximum energy saving and maximum extra energy cost for our robots is around 13%.

Line 473: “In our robotic experiments we find that a follower can save energy in any relative spatial position with respect to a near neighbour if it adopts a specific relative tailbeat phase difference, and that this depends predominantly on the front-back distance (Supplementary Figs. 9 and 10). The maximum energy saving of the follower robot was found to be lower than many previous predictions from theoretical models^{3,52} or numerical simulations^{28,31}. This might be related to turbulence in the flow at high Reynolds number, and/or due to the effects of viscosity and the three dimensional nature of the flow in real systems. For example, Verma et. al¹⁰ found that the power costs of the follower can decrease up to 36% in 2D but only 5% in 3D CFD simulations.”

R1.Q15:

And why then do fish seem to show a preference for certain interactions with flows (i.e. Fig. 3)?

R1.A15:

We added several hypotheses regarding this point. These include:

1. The real fish system might reap greater overall benefits, relative to costs, as there are inevitable differences between the robotic fish system and real fish system, the latter being much more efficient (R1.A13).
2. The benefit might come for free by passive reaction to flow properties, a hypothesis also mentioned by Reviewer #1 (R1.A22).

R1.Q16:

5) It is not at all clear if the actual fish display phase locking that is quantitatively similar to the phase locking that is found to be energetically favorable in the robotic fish system. Here the presentation is vague and should be made direct and clear, and the result should be more completely interpreted. In particular, it seems from Fig. 2A that the spatial phase $\phi_0 = 0.3\pi$ is optimal and yet from Fig. 3 that fish adopt $\phi_0 = -0.2\pi$. The difference of about 0.5π seems significant given that a change of ϕ by π changes the savings from most optimal to most detrimental (energy losses). In any case, I think an open, honest and complete discussion would help the

reader make sense of the results. Do actual fish adopt the optimal spatial phase? If not, what are the savings found in the robotic system for the spatial phase that the actual fish do adopt?

R1.A16:

The initial phase differences (spatial phase difference in the “Lagrangian specification”) are different between the robotic fish case and real fish case. The reasons are:

1, The systems are different. Although we used bio-inspired robotic fish as a model to explore the nature and consequences of hydrodynamic interactions in pairs of fish, we still have limitations for these studies. For example, the full degrees of freedom possible in undulations and body kinematics of real fish and the three-dimensional movements cannot be fully represented in (any contemporary) robotic fish experiments.

2, Φ_0 in the robotic fish system is optimal (Fig. 2 and its caption: “ Φ_0^ is the optimal initial phase difference (fitted to the data points of maximum energy saving, ...)”) and in the real fish system is slightly sub-optimal (Line 515: “We also find fish adopted VPM with a slightly sub-optimal initial phase difference ($\Phi_0 = -0.2\pi$) instead of global optimal initial phase difference ($\Phi_0 \approx 0$, Fig. 4). While this difference was found to have only very small consequences regarding energy savings, it is potentially an indication that fish must consider more than saving energy while swimming together^{15,53}, and that moving in order to acquire salient sensory information (in order to better detect food, possible threats, changes in swimming direction of neighbours⁵⁴) will typically result in deviations from what is hydrodynamically optimal. That is, real fish have to balance many selective pressures when making movement decisions.”).*

Overall, the deviation from perfectly optimal behaviour for one feature of their lives (here, energy saving) is not unexpected in a real biological system where may selection pressures - such as the requirement to move to obtain salient information from conspecifics and the external environment is also extremely important.

R1.Q17:

Can any differences in results be explained by differences in the two systems?

R1.A17:

As explained above (**R1.A16**), the difference in the initial phase difference Φ_0 might come from the system differences. We revised this in the main text as:

Line 411: “We note that unlike in the robotic fish experiments, where we can directly measure power consumption, such measurement is not possible for the real fish. We cannot directly apply the measured costs and benefits of swimming together from our bio-mimetic robots, since fish propulsion via muscle is far more efficient than by motors⁴⁹, their propulsive undulations travel more smoothly, their skin is coated with mucous to decrease resistance to fluids⁵⁰ etc. Thus real fish are likely to obtain considerably greater benefits if they employ VPM than do our robots.”

R1.Q18:

How do the authors interpret these findings for schooling more generally?

R1.A18:

We agree with Reviewer #1 that this point is very interesting and important. As we point out in the discussions (Line 525: “In the future it will be valuable to investigate more complex social scenarios (going beyond pairs of fish).”). While extremely difficult it will be a natural next step in future work. Based on previous studies, fish swim dynamically in pairs in schools (Line 181: “We note that this is biologically meaningful as swimming in pairs is both the most common configuration found in natural fish populations^{6,9,36,37}, and it has been found that even in schools fish tend to swim close to only a single neighbour^{6,36}.”). It is critical to understand how the energy savings first in pairs, and then in schools. In addition, for our biological experiment, we choose a fish species (goldfish, *Carassius auratus*) mainly for practical reasons but concentrating on that it has a similar Reynolds number and natural swim speeds as the robot. But no further selection criteria was applied as, if important biologically, it should be common among species. However, for species that typically swim in fast streams and/or in more energetically demanding/challenging conditions (e.g. eels migrate up to 6000 km, Schmidt, J., 1923. *Breeding Places and Migrations of the Eel*. *Nature*, 111(2776), pp.51–54.) we may well find even stronger VPM. Further studies would be needed to clarify this.

R1.Q19:

Some more minor comments:

1) Title: Have the authors really shown this? It seems like an overstatement given that no energy measurements on fish are performed. I think a more honest title conveying more directly what is actually shown in the work would be more useful and still be impactful.

R1.A19:

We revised the title as “Fish can save energy by vortex phase matching with a near neighbour”. We would be willing to accept suggestions for alternative titles, however.

In the subsection of “Testing our predictions with real fish”, we clarified the way we analysed the real fish data and how we draw the conclusion that fish dynamically adapt vortex phase matching and how this behaviour can result in them saving energy.

The energy cost is evaluated by measuring the tailbeat frequency and amplitude ---- we find that the kinematics of our fish, in the energy saving regime are in accordance with previous studies (Liao, J. C., Beal, D. N., Lauder, G. V. & Triantafyllou, M. S. Fish exploiting vortices decrease muscle activity. Science 302, 1566–1569 (2003)). We also clarify that we are not measuring the absolute power cost and measuring the relative power cost.

Line 404: “We also estimated the “power consumption” of the follower fish (by approximating it from the measured amplitude and frequency of its body undulation⁴⁶) under different types of hydrodynamic interactions. Since fish in the flow tank tend to match their swim speed to the flow speed of the water (see Supplementary Fig. 25), here we are not considering absolute energy savings (which would be maximised by not swimming at all), but rather the relative power consumption for a given swim speed in the presence, or absence, of hydrodynamic interactions. We note that unlike in the robotic fish experiments, where we can directly measure power consumption, such measurement is not possible for the real fish. We cannot directly apply the measured costs and benefits of swimming together from our bio-mimetic robots, since fish propulsion via muscle is far more efficient than by motors⁴⁹, their propulsive undulations travel more smoothly, their skin is coated with mucous to decrease resistance to fluids⁵⁰ etc. Thus real fish are likely to obtain considerably greater benefits if they employ VPM than do our robots.”

Therefore, in this paper, we are able to see fish do behave in a way that will save energy, and as we write in the main text-Line 451: "... we can estimate that by behaving as they do, fish (in our flow conditions) save (by accumulating all benefits and extra costs; where a random behaviour would give 0) an overall 15% of the total possible (which would be achieved by perfectly adopting the optimal phase to the neighbour-generated vortices at all the time, Supplementary Fig. 28)."- but we are not yet able to tell how much energy, in absolute terms, the fish will save (we are not aware of any known experimental technique that would allow us to directly do so, as discussed in the Introduction).

R1.Q20:

2) Abstract: In addition to the points given in the Major comments above, it seems the focus on the two-fish situation should be stated. Also, there does not seem to be any broader interpretation of the results for fish schools.

R1.A20:

We agree and have clarified this in the abstract and in the body of the paper that we are mainly focusing on the pairs of fish and that is biologically meaningful for species of schooling fish.

*In the abstract-Line25: "...To address this problem, we first investigate hydrodynamic interactions between two biomimetic fish-like robots, allowing us to directly measure energy consumption associated with swimming together. ... Experiments with pairs of freely-swimming goldfish (*Carassius auratus*) reveal that, ..."*

In the body of the paper-Line 179: "...Here, due to the complexity of the problem (as discussed above) we consider hydrodynamic interactions between pairs of fish. We note that this is biologically meaningful as swimming in pairs is both the most common configuration found in natural fish populations^{6,9,36,37}, and it has been found that even in schools fish tend to swim close to only a single neighbour^{6,36}. ..."

R1.Q21:

3) Introduction: The background review is quite detailed, but one common thread in many recent papers seems to be left out. Namely, many papers from the last ~5 years have shown that specific phase relationships can come about entirely passively from the flow interactions: Zhang PRL 2015, Ramananarivo PRF 2017, Newbolt PNAS 2019 and others.

R1.A21:

We thank you for the suggestion and apologise for this unintentional omission. We have now included this possibility (Line 96) and added these references. We revised as:

Line 96: "Recent experiments employing simplified mechanical models^{11,29,30}, as well as theoretical studies^{23,28}, have also found that active swimmers could passively converge to specific stable spatial formations to save energy as a result of hydrodynamic interactions."

R1.Q22:

This represents an alternative to the view that specific phases or positions must be achieved through active sensing-and-response behaviors. In fact, it seems that the authors' observations on actual fish might be well explained by this hypothesis, especially given that fish display vortex locking even without vision and without the lateral line.

R1.A22:

We agree with the Reviewer. While know that real fish do not adopt specific spatial positions (see Line 99: "We note, however, that experiments with real fish find that individuals do not adopt any such specific spatial configurations⁵⁻⁷.")) our work certainly provides evidence that passive reflex mechanisms (i.e. mechanical feedback) may be a very important factor, as may be passive or sensing via proprioception. We agree that this differs from the active sensing (in which animals move specifically in order to obtain information) view of how this behaviour could be coordinated. We feel that this is, in itself, a useful biological insight - especially that the lateral line need not be necessary - and could help guide future design for robotics. We discussed this in the Main text as:

Line 506: "...In addition to this we find that neither a functioning lateral line or visual system is required to regulate accurately inter-individual hydrodynamic interactions suggesting that fish might be able to save energy passively^{11,28,29} and/or by utilising other sensors (such as proprioceptive sensors⁴⁶⁻⁴⁸). ... Further studies on real fish should be conducted to test these hypotheses."

R1.Q23:

In general throughout the paper, the authors should consider more discussion along these lines. For example, if fish adopt a different phase from what is optimal, then could it be that they adopt a phase that is passively preferred due to flow interactions?

R1.A23:

Yes, we have now done so in the revised manuscript (also see R1.A21 and R1.A22)

R1.Q24:

4) The very recent work by Oza PRX 2019 also seems highly relevant, especially in regard to different schooling configurations.

R1.A24:

Thanks for the recommendation, we cite this paper in the new version where we introduce the relationship between formations and energy saving (see Ref 23). For example, we included this in the following sentence:

Line 96: "Recent experiments employing simplified mechanical models^{11,29,30}, as well as theoretical studies^{23,28}, have also found that active swimmers could passively converge to specific stable spatial formations to save energy as a result of hydrodynamic interactions."

R1.Q25:

5) Page 6 first sentence of Results section: "costs" might be replaced with "benefits" or, more neutrally, "costs and benefits"

R1.A25:

Yes, we agree. We have revised "costs" to be "costs and benefits" in the new version.

R1.Q26:

6) Page 6 near bottom: "swimming speed/flow speed" should be "swimming or flow speed"

R1.A26:

Yes. This is revised accordingly in the new version.

R1.Q27:

7) The fact that the robots are fixed in a flow and not free swimming should be made clear. Given this, it seems that the robots are not in mechanical equilibrium in general, and so the conditions explored in the experiments would not be realized by free swimmers.

R1.A27:

We have now addressed this issue directly, including adding a new analysis, incorporated in the main text as:

Line 196: “To establish whether the robotic fish connected with a thin bar has similar hydrodynamics compared to when free swimming, we measured the net force (of the drag and thrust generated by the fish body in the front-back direction) acting on the robot in the flow tank. The measured net force over a full cycle (body undulation) was found to be zero; thus the bar is not measurably impacting the hydrodynamics of our robot fish in the front-back direction as they swim in the flow tank (Supplementary Fig. 7).”

Furthermore we have now conducted a computational fluid dynamics (CFD) simulation, also incorporated into the main text:

Line 204: “To further validate the utility of the platform, we also compared the power consumption of our robots swimming side-by-side, for different relative phase differences Φ , with equivalent measurements made with a simple 2D computational fluid dynamics (CFD) model of the same scenario. In both cases (see Supplementary Fig. 8a and c for robotic experiments and CFD simulations, respectively) we find that there exists an approximately sinusoidal relationship between power costs and phase difference which is defined as $\Phi = \phi_{\text{leader}} - \phi_{\text{follower}}$ (Fig. 1c and d). Due to the 2D nature of the simulation, as well as many other inevitable differences between simulations and real world mechanics, the absolute power costs are different from those measured for the robots, but nevertheless the results from these two approaches are broadly comparable and produce qualitatively similar relative power distributions when varying the phase difference between the leader and follower. These results indicate that our robotic fish are both an efficient (making estimates of swimming costs is far quicker with our robotic platform than it is with CFD simulations) and effective (in that they capture the essential hydrodynamic interactions as well as naturally incorporate 3D factors) platform for generating testable hypotheses regarding hydrodynamic interactions in pairs of fish.”

R1.Q28:

8) The quantity called efficiency is not a true efficiency but more like a dimensionless power consumption, normalized by that of a single swimmer. The authors may consider a different name.

R1.A28:

Yes. We have revised to be “a dimensionless relative power coefficient” and subsequently described (more simply) as “a relative power coefficient”

R1.Q29:

9) Sentences spanning pages 7 and 8 on range of interaction: This seems like an overstatement given that the breakdown of a wake depends sensitively on aspects such as flapping kinematics and tail geometry. I would suggest refraining from making general claims based on one set of such parameters.

R1.A29:

We agree. This sentence has been removed.

R1.Q30:

10) Page 8, reference to figures 7 and 8: These are not in the main paper, so perhaps a citation to the supplement is intended here?

R1.A30:

Yes, we corrected this to refer to the relevant supplementary figures.

R1.Q31:

11) Fig. 1 seems to emphasize flow visualization, which is not at all used in anyway throughout the rest of the work.

R1.A31:

We agree. Fig. 1 has been redrawn with emphasis on the variables of front-back distance, left-right distance and phase difference, and an example of the tests of power cost relative to phase difference at specific formation (left-right distance $G=0.27$ BL and front-back distance $D=0.33$ BL). We keep an example of the flow visualisation as we think it is helpful to an interested reader who may be unfamiliar with the vortices generated during fish swimming.

R1.Q32:

Missing is any information about the actual kinematics, space of configurations explored, sample power traces, etc. These are much more important data for the plots to come.

R1.A32:

Fig. 1 has been redrawn with more introductions on the variables we studied and a sample of the power traces. The analysis of kinematics has been enhanced by adding a supplementary Fig. 2. The space of configurations explored are shown in Fig. 1b and explained in its caption.

R1.Q33:

12) Fig. 2C and caption: As in other parts of the paper, it seems spatial phase could tighten up the explanation here. It seems the authors are saying something simple about spatial phase, but I was not able to grasp the point of the schematics and very long caption.

R1.A33:

As discussed above (R1.A7), we have now made clear that the results can be considered in either framework. Although the “spatial phase” might be easy to comprehend for some readers, audiences from other fields (e.g. biology background) may find this term difficult to understand. Secondly, we described the system following the concept of “Eulerian specification”, it will be more reasonable to use “Phi_0” (phase difference at front-back distance 0) in our case.

R1.Q34:

13) Fig. 3: The plots (together with vague text) are a little misleading, given the colorbar runs over just a few percent in terms of occupancy probability (if this is in fact what is meant by the quantity P).

R1.A34:

We have updated this figure and the caption with more details:

“... Occurrences of observed swimming relationships (probability density function) as a function of phase difference Φ and front-back distance D between leader and follower illustrated by colour-coding...”

R1.Q35:

14) The plots of Figs. 2 and 3 should be brought into alignment in terms of axes range and such. For example, D should go from 0 to 1 in these plots (leaving blank $D < 0.22$ if this was not explored in Fig. 2) and ϕ from 0 to

2*pi. The repetition of the same data in Fig. 3 for phi outside this range seems strange and not useful. Uniformity of the plots is important to, for example, compare the savings bands of Fig. 2 with the occupancy bands of Fig. 3.

R1.A35:

Thanks for pointing this out. We have updated the two figures with the same range of D and kept only one replicate for the real fish data. We think that the repetition is still important here, as while the phase is periodic, the linear trend passes through multiple periods.

As with typical biological data, we have variation from multiple sources, and we discuss this in the revised version:

Line 458: “ However we would never expect fish motion to be completely dominated by a need to save energy as they must also move in ways as to obtain salient social and asocial information from their visual, olfactory, acoustic and hydrodynamic environment, such as to better detect food⁵¹, environmental gradients⁴² and threats¹⁵. ...”

Despite this, fish preferentially use the phases given the VPM rule, and we do not want to just demonstrate this via statistical analysis, but also to show this visually (as raw data) as well. We believe that the presentation spanning through two periods convey a clearer message for the readers.

In order to fulfill the request of the Reviewer in comparing directly the real fish data and robotic fish data, we added a Supplementary Fig 27.

R1.Q36:

15) I found the term “socially-generated flow” to be odd since the source is physical and not social or behavioral – “collectively generated”?

R1.A36:

We see the ambiguity. We had intended it to mean generated by another conspecific. To disambiguate this we have revised this to be “neighbour-generated flow”

R1.Q37:

16) Page 13, reference to reflex or proprioceptive response – couldn't all of these data also be explained by the simple hypothesis that phase locking

comes about passively/physically through flow interactions? See minor point 3 above.

R1.A37:

Please see our response to R1.A21 and R1.A22 above.

R1.Q38:

17) I didn't understand the reasoning regarding Fig. 4. These data seem to have no obvious connection to energy savings, yet it is claimed (in the caption and text) that "fish save energy". I think this portion of the paper needs to be carefully reworked in terms of what is trying to be conveyed. The direct message seems to be that fish have different flapping frequency and amplitude when in different spatial phase relationships. But what more can be said? Also, these changes measured are very slight – do they matter? Again, more discussion about quantitative results and their interpretation is needed.

R1.A38:

In this analysis we provide an estimation of power cost by analysing tailbeat frequency and amplitude as a proxy for energy expenditure (since the latter cannot be directly measured) (also see R1.A19) following several other studies (e.g. Liao, J.C. et al., 2003. Fish exploiting vortices decrease muscle activity. Science, 302(5650), pp.1566–1569.) from the literature. From Fig. 3, we know fish choose the preferred phase difference linearly depending on the front-back distance, which can be explained by our vortex phase matching. However, we do not know if the preferred initial phase difference (spatial phase difference) provides energy saving benefits. Therefore, we conducted analysis and showed the results in Fig. 4, where we estimate and compare power cost by frequency and amplitude between different initial phase differences Φ_0 and $\Phi_0 + \pi$. If Φ_0 results to a lower frequency and a higher amplitude compared to $\Phi_0 + \pi$, then Φ_0 leads to relatively energy saving.

From Fig. 4, the direct message is that fish using VPM with different Φ_0 will statistically result in different frequencies and amplitudes, indicating Φ_0 determines the power cost of the fish. And the most used phase difference gives $\Phi_0 = -0.2\pi$ is located well within the energy saving zone. We think that this is useful information for the reader.

The variation of the frequency relative to average is around 4.3% and amplitude relative to average is around 5.6%, which broadly correspond to the energy saving ratio found in the robotic fish experiment.

R1.Q39:

18) First sentence of Conclusions: “costs” might be replaced with “benefits” or, more neutrally, “costs and benefits”

R1.A39:

Adopted.

R1.Q40:

19) Methods section: This does not seem to be about methods but reads like a summary of the paper.

R1.A40:

We changed the summary to be detailed methods.

Reviewer #2 (Remarks to the Author):

R2.Q1:

Referee Liang li and others, schooling Fish save energy by vortex phase matching

This is an interesting paper in which energy loss is measured for pairs of robotic fish. Results are used to develop a mathematical model of matching the phase of vortices in order to save energy when swimming in a pair. Predictions of this model are subsequently tested in pairs of real fish and confirmed.

This method of experiments with robotic fish, mathematical model and empirical testing is very good. The paper needs improved in that it should state more clearly already in the abstract, and also throughout the main paper, that results merely concern pairs of fish not schools.

The main paper should also be more clear regarding:

1. What is meant with constructive and destructive reaction to vortices by the follower, see for instance, caption of figure 2 c-e.

R2.A1:

“Constructive reaction” means the follower interacts with the vortex by flapping its tail in the same direction of the induced flow.

“Destructive reaction” means the follower does the opposite way. In the revised paper, the constructive reaction is explained in the main text and we refer to figure (Fig. 2c-e) where the concept is illustrated. (Line 286: “... reveal that the high-efficiency regions (indicated by the blue colour in Fig. 2a) correspond with highly-constructive interactions between shed vortices, and thus that the observed effects are driven by the tail flapping direction and the induced flow direction (Fig. 2d and e, Supplementary Fig. 6, Movie 3. ”).

R2.Q2:

2. As regards details of phase matching of vortices, in the last paragraph of page 7, it is mentioned that the benefits of swimming together decay fast with increasing distance of follower to leader, figure 2A. It needs more explanation how we can observe this in fig 2A.

R2.A2:

We agree. A simple figure catching the peak of energy saving as a function of the distance is now given in Supplementary Fig. 11. From this figure, we can see a fast decrease regarding the maximum energy saving.

R2.Q3:

It should be explained what the efficiency coefficient refers to and what positive and negative values imply.

R2.A3:

The “efficiency coefficient” (now named “Dimensionless relative power coefficient”, and more simply “Relative power coefficient”, following Reviewer #1’s suggestion (R1.A28)) is defined according to Eq. (1) in the main text. This evaluates the benefits or costs relative to swimming alone. Positive values indicate relative energy saving and negative values indicate relative power cost. This is revised in the new version (Line 246: “Positive values (blue in Fig. 2a) and negative values (red in Fig. 2a) respectively represent energy saving and energy cost relative to swimming alone.”).

R2.Q4:

Also what is meant in fig 2 CDE should be explained in ore detail.

R2.A4:

We agree. We revised Fig. 2 and added extra information (and attempted to express it more clearly than before) to show the principle that fish can save energy by adjusting body phase regardless of the front-back distance. The reason is that fish interact with the vortex shed by the leader similarly (Phase differences between vortex and follower are the same Fig. 2d, e). All these are revised in the new version (Line 290: "Thus energy savings seem predominantly determined by macroscopic properties of the flow, and specifically the follower's reaction to the primary induced flow of the coupled reverse K'arm'an vortex (Fig. 2c-e, Supplementary Fig. 6) produced by the leader.")

R2.Q5:

3. There is no mention of the size of the tank relative to the robotic fish and real fish, but this may have a large effect on flow. For instance first paragraph on page 7 of main text should inform on this and on the effects of walls on flow.

R2.A5:

We apologise and have now moved this information from supplementary information to the main text (Line 185: "To evaluate the energetics of swimming together we conducted experiments on our pair of robotic fish in a flow tank (test area: 0.4m-wide, 1m-long, 0.45m-deep)..."). Moreover, we have now added more details by including experiments and data analysis to show that the effects of the boundary in our experiments are small. One typical experiment is that we measured the net force of the robotic swimming in the flow tank given the flow speed measured by free swimming robotic fish with the same kinematics (also see R1.A27).

R2.Q6:

4. The relation between energy saving zone in figure 4 and vicinity of vortices in text second paragraph below figure 4 should be made clear.

R2.A6:

We agree. A more detailed introduction of Fig. 4 is now provided in the main text. See a paragraph starting from Line 431: ("To further test if fish can save energy by adopting VPM with the typical vortex-body hydrodynamic interactions ($\Phi_0 = -0.2\pi$), we compared an estimation of the power consumption" under different hydrodynamic interactions. Since the hydrodynamic interactions are mainly determined by the initial phase

difference Φ_0 (see above), we analysed performance in the full possible range from $-\pi$ to π (see Supplementary Fig. 26 for the detailed method).”).

*In Fig. 4, we show how the “power cost” is related to the initial phase difference Φ_0 (also see **R1.A38**). From this figure, we see the initial phase difference Φ_0 at the peak of the amplitude and valley of the frequency is the optimal initial phase difference. This corresponds to the optimal way for the follower to interact the vortex if saving energy were the only criterion (Fig. 2c, d). The real fish employed an initial phase difference (dashed line in Fig. 4) close to the optimal value, and well within the energy saving regime; therefore the way fish interact with the vortex is closer to the way described in Fig. 2c, d but with a small shift.*

R2.Q7:

The manuscript needs also improved as to its references, sometimes pairs are cited that have no relation to the text, for instance, in introduction page 3, in the second paragraph reference 14 is incorrectly located, e.g., page 4 last paragraph Boschitsch et al is mentioned with an nonmatching reference to 23.

The authors need to go carefully through the complete text to amend references.

R2.A7:

We checked the references and the list at the end of the main text was correct. But for the SI, we had a separate reference list (also with a numbering starting from 1), and the main text and the supplementary information (with both their reference lists) were all in one pdf file. We apologise as we think this may have caused confusion. In the updated version, the numbering of the references appears in the SI starts at where the reference numbering finished in the main text.

Reviewer #3 (Remarks to the Author):

R3.Q1:

This is the review of NCOMMS-19-363561 titled “Schooling fish save energy by vortex phase matching”. In this manuscript the authors investigate the energy saving mechanisms of fish schooling through experiments with robots and real fish. In the robotic fish experiments, a pair of fish are placed at different lateral and front-back space D with different phase angles while

the energy cost is measured. It was found that the power was more sensitive to the front-back spacing and phase angle than the lateral spacing. The length of the fish, tail beat amplitude, and frequency were kept constant. It was found that the phase difference proportional to the fD/u minimized energy consumption. It was suggested that the mechanism for energy reduction is constructive vortex interaction. This idea was also tested on a small number of goldfish pairs. The phase of the goldfish pairs and their distance D , qualitatively agreed the robotic experiments (not as good though).

R3.A1:

Thank you for spending time reviewing our paper. We have revised our paper to make it more clear and have added more experiments to justify why the identification and testing of our rule is important, and have included a more comprehensive discussion of how our findings contribute to addressing a very long-lasting biological problem of relevance to both biology and engineering. In particular we have stressed the importance of creating (here using the robotic fish and the development of a biologically-plausible rule) and testing our predictions with real fish.

Please also see also our answer at R1.A6. Briefly, we used robotic fish to generate a hypothesis and then test this hypothesis in the real fish system. Therefore, we did not expect to get an exact match between the robotic fish data and real fish data. We have now clarified this, and why this must be the case, in the main text:

Line 137: "Our focus here is twofold; firstly we aim to develop a unified understanding of the previously-described hypotheses regarding how real fish should be expected to behave if they are exploiting hydrodynamic interactions (i.e. to reveal the behavioural rule(s) that fish would be expected to adopt to achieve benefits), and secondly, and most importantly, we seek to test our predictions in experiments conducted with real fish."

And

Line 411: "We note that unlike in the robotic fish experiments, where we can directly measure power consumption, such measurement is not possible for the real fish. We cannot directly apply the measured costs and benefits of swimming together from our bio-mimetic robots, since fish propulsion via muscle is far more efficient than by motors⁴⁹, their propulsive undulations

travel more smoothly, their skin is coated with mucous to decrease resistance to fluids⁵⁰ etc. Thus real fish are likely to obtain considerably greater benefits if they employ VPM than do our robots.”

*Further discussion of this issue can be found in **R1.A6**, above.*

R3.Q2:

I do not recommend this paper for publication because of the following reasons:

R3.A2:

We are sorry about Reviewer #3’s negative evaluation but we would like to point out that we believe that Reviewer #3 came to this conclusion because they interpreted incorrectly some parts of our work. We feel also responsible if such misinterpretation occurred, and thus we revised several parts to be able to more clearly state our concepts and findings. We show later in more details point by point. Other points in Reviewer #3’s list refer to actual weaknesses of the previous version (these have now been addressed, see above). We thank the reviewer for those comments, we consider them with great attention. They helped us to greatly improve our manuscript.

R3.Q3:

1) The interaction is only investigated in pairs rather than in a school. So the title can be misleading.

R3.A3:

*We agree. We had meant “schooling fish” to indicate species of fish that can (but do not always) school, as this is how such species are referred to in the biology literature. However, due to the ambiguity we have changed the title to simply indicate “fish” and we have modified both the title and the abstract to make more explicit that our studies focus on pairs of fish (also see **R1.A19** and **R1.A20**).*

We also discuss this issue in the main text:

Line 179: “...Here, due to the complexity of the problem (as discussed above) we consider hydrodynamic interactions between pairs of fish. We note that this is biologically meaningful as swimming in pairs is both the most common configuration found in natural fish populations^{6,9,36,37}, and it has

been found that even in schools fish tend to swim close to only a single neighbour^{6,36}. ...”

R3.Q4:

2) I have a few concern regarding the vortex phase matching hypothesis: a) it does not consider the lateral movement of the vortex. In fact, for constructive interaction, the location of the similar-sign vortices should be similar both in axial and lateral locations;

R3.A4:

Given this and later comments (for example, R3.Q9), we suspect that Reviewer #3 is considering vortex-vortex interactions. But our paper considers instead short distance vortex-body interactions. Namely, the follower’s body interacts with the vortex shed by the leader. Due to this misunderstanding we have now made much more explicit how and why we do so. For example:

Line 261: “Although we know fish generate reverse K’arm’an vortices at the Reynolds number ($Re = Lu/\nu \approx 10^5$, where L is the fish body length, u is the swimming or flow speed and ν is the kinematic viscosity) in our experiments³⁸ (Supplementary Movie 1), turbulence will dominate over longer distances¹⁷. In accordance with this, we see a relatively fast decay in the benefits of swimming together as a function of D (e.g., $D > \sim 0.7 BL$, see Supplementary Fig. 11 for the relationship between the maximum relative power coefficient and the front-back distance), a feature we also expect to be apparent in natural fish schools (where it would likely be exacerbated by what would almost always be less-laminar flow conditions). Therefore, we expect, based on our results, that hydrodynamic interactions are dominated by short-distance vortex-body interactions (with $D < 0.7 BL$). While complementary to previous studies of vortex-vortex interactions^{3,16,31}, here we focus on vortex-body interactions since these are expected, based on the above results, to have a far greater impact on energetics when swimming (for this regime of Reynolds number)”

We are very sorry about this misunderstanding. And we revised carefully our manuscript keeping in mind how to avoid this misinterpretation. We believe this made our manuscript clearer and stronger. We thank Reviewer #3 for bringing this issue to our attention.

R3.Q5:

b) It is only tested for one velocity and frequency (u and f), i.e., out of the three parameters fD/u only one is varied. The dependence at other velocities and frequencies is not shown;

R3.A5:

We thank Reviewer #3 for making this point. We have indeed conducted our main robotic fish experiments with various frequencies f . We now include these results and show them in Supplementary Fig. 13. This shows a positive correlation between the optimal phase difference and frequency f , in agreement with the functional relationship considered in the VPM rule.

The flow speed u was also varied systematically in the experiment with real fish. Data on this are shown in Supplementary Fig. 26. The validity of the VPM rule can be clearly detected there.

In the previous version of the manuscript we did not include all our experiments mainly because we were trying to keep the paper as brief and as accessible as possible. Also, as mentioned in other answers, some of our previous work we only included in the SI where it was likely hard to find - we apologise, and have included this work now.

R3.Q6:

and c) the speed of vortices is taken to be similar to the swimming speed. This has not been justified. Because of these, I do not feel the vortex phase matching is adequately tested.

R3.A6:

The rule we proposed is a deliberately minimal model that could be used for generating and testing predictions. The value of this approach is that it makes the main mechanism easy to understand, and most generally applicable. The utility of the assumptions is whether or not such a simple model can provide a mechanistic explanation for experimental data. In our case we provide strong evidence that it can (see, for example Fig. 2a-b, where the model predictions are compared to direct tests conducted with the robotic fish platform). Generally speaking the simplest possible model that can well explain a real phenomenon, and can make testable predictions, which are then validated, is a powerful approach. The reason this assumption that the speed of vortex equals the flow speed in the flow tunnel provides such a good match to our experimental data is that we consider local vortex-body interactions (see also R3.A4), which our robotic

experiments show to be the dominant factor (this would also be expected to be especially true for real biological systems where there are many sources of sensory and environmental noise). The validity of our assumption is justified by the good fit between the derived VPM rule and the experimental results of both the robotic and real fish experiment in case of such relatively small front-back distances.

We have modified the main text to explain the utility of minimal models and why such a simple model is appropriate for our study (the value being that it works):

Line 304: “The main assumption is that the flow is inviscid, and thus the vortex moves backward with constant morphology and velocity. This is reasonable in our case since, guided by our robotic results, we focus on the direction of the induced flow of the reverse Kármán vortices and short-range hydrodynamic interactions between the vortex and the body.”

From a biological perspective it is also important to develop and test predictions in conditions that approximate typical conditions in which the organisms find themselves in nature. For example, we could have explored whether fish utilise neighbour-generated vortices under conditions of more extreme flow (well above preferred swimming speed). This would maximise our chances of finding a result, but may not make our work relevant to the typical conditions in which these organisms find themselves. Thus we employed a more challenging, conservative approach - asking whether in the daily lives of fish this behaviour is relevant, as opposed to whether it can be relevant only under high flow conditions. We have now included a summary of this in the main text - Line 366: “Fish were placed in a flow tank with uniform flow speed U , with U ranging from 1.2 to 1.6 BL/s (equivalent to their natural swimming speed⁴⁴) with increments of 0.1 BL/s , and filmed with a camera at 100 frames-per-second (Supplementary Fig. 15 and Methods) from which the body posture of each fish was estimated automatically using DeepPoseKit⁴⁵ (Fig. 3a, Supplementary Fig. 16 and Movie 4). We analysed all data where leader-follower pairs were spatially positioned such that the follower could potentially interact with the vortices shed by the leader (see above), up to a separation of 1 BL in front-back distance (801,000 frames; see Supplementary Information for details).”

In our robotic fish experiments, we first conducted experiments with robotic fish to measure the swimming speed, and the flow speed in the flow tank

was set as the swimming speed. A net force test was conducted to show the balance between the thrust and drag caused by the flow in the front-back direction (also see R1.A27). This indicates that the flow speed can be evaluated by the free swimming speed. In a free swimming case the vortices do not move compared to the global reference/medium, as depicted on Supplementary Movie 1.

R3.Q7:

3) It is not clear if the constructive vortex interaction occurs at all time instants during a tail beat or just at specific instants. If it happens at all time instants, it can also increase the drag (in addition to thrust). There are two force peaks per tail beat corresponding to back and forth motion of the tail in each tail beat. If constructive interaction occurs, it can increase maximums as well as the minimums during a cycle.

R3.A7:

Theoretically, constructive vortex interactions can happen continuously in time. For example, when the leader and follower have specific formation and constant phase difference, the follower will continuously save energy in each period. Because of the pre-set parameters for the robotic fish, this can occur for arbitrarily long periods of time. For the real fish, most typically there are no continuous stable spatial formations, thus fish swim dynamically with changing relative positions and tailbeat frequencies. As a consequence of this, as we have revealed here, they continuously change relative phase (in the way that we predicted they should via our identified VPM rule) to save energy (Main Fig. 3). This indicates that our VPM rule is robust, and is employed by real fish.

R3.Q8:

4) My main concern regarding the results of the robotic experiments is that the robotic fish are not self-propelled. In fact, it is not clear if the average force over the fish are zero over the cycle. The energy saving during free swimming and tethered conditions might not be the same.

R3.A8:

We thank you for making this point. A similar concern was raised by Reviewer #1 (also see R1.A27). We apologise for not having included the analysis we had conducted to evaluate this. It is now included in the main text:

Line 196: “To establish whether the robotic fish connected with a thin bar has similar hydrodynamics compared to when free swimming, we measured the net force (of the drag and thrust generated by the fish body in the front-back direction) acting on the robot in the flow tank. The measured net force over a full cycle (body undulation) was found to be zero; thus the bar is not measurably impacting the hydrodynamics of our robot fish in the front-back direction as they swim in the flow tank (Supplementary Fig. 7).”

and

Line 204: “To further validate the utility of the platform, we also compared the power consumption of our robots swimming side-by-side, for different relative phase differences Φ , with equivalent measurements made with a simple 2D computational fluid dynamics (CFD) model of the same scenario. In both cases (see Supplementary Fig. 8a and c for robotic experiments and CFD simulations, respectively) we find that there exists an approximately sinusoidal relationship between power costs and phase difference which is defined as $\Phi = \phi_{\text{leader}} - \phi_{\text{follower}}$ (Fig. 1c and d). Due to the 2D nature of the simulation, as well as many other inevitable differences between simulations and real world mechanics, the absolute power costs are different from those measured for the robots, but nevertheless the results from these two approaches are broadly comparable and produce qualitatively similar relative power distributions when varying the phase difference between the leader and follower. These results indicate that our robotic fish are both an efficient (making estimates of swimming costs is far quicker with our robotic platform than it is with CFD simulations) and effective (in that they capture the essential hydrodynamic interactions as well as naturally incorporate 3D factors) platform for generating testable hypotheses regarding hydrodynamic interactions in pairs of fish.”

R3.Q9:

5) The mechanism described does not clearly show the constructive interaction of vortices. In fact, the main figure 2 and 5 (supplementary) does not show the vortices of the follower fish. This makes me wonder if the mechanism observed by the authors is due to vortex interaction. As an example, I can provide another hypothesis on the energy saving based on the same figure: the main portion that generate thrust in such fish is the tail. The follower is positioned such that the induced flow in the wake of the leader increases the angle of attack to the tail of the follower and, consequently, increase the generated thrust.

R3.A9:

We think this results from a misunderstanding. We are not considering vortex-vortex interactions, but rather (as explained above) vortex-body interactions. This has now been further emphasized in the main text, Line 270 (“Therefore, we expect, based on our results, that hydrodynamic interactions are dominated by short-distance vortex-body interactions (with $D < 0.7 BL$). While complementary to previous studies of vortex-vortex interactions^{3,16,31}, here we focus on vortex-body interactions since these are expected, based on the above results, to have a far greater impact on energetics when swimming (for this regime of Reynolds number”). As the Reviewer suggests for such interactions the angle of attack to the tail (the angle difference between the induced flow and the tail flapping direction, emphasized in the main text in Line 288(“...and thus that the observed effects are driven by the tail flapping direction and the induced flow direction (Fig. 2d and e, Supplementary Fig. 6, Movie 3).”)) is key to the benefits or costs to the follower. It determines how the fish body interacts with the vortex to save energy or increase the thrust.

R3.Q10:

6) The flow visualization of interaction of the fish using hydrogen bubbles show instabilities in the flow which does not seem realistic for actual fish probably because of the tethered nature of the fish. It is very different than the reverse von karman for a single fish.

R3.A10:

The camera view of hydrogen bubble visualizations are small (see Supplementary Fig. 6a, the shaded area). And we wanted to visualize the vortex-body interaction closely. Therefore, we are only able to show partial vortex and partial fish body (Supplementary Fig. 6b and Movie 3). The full Reverse Karman Vortex shed by the robotic fish is now shown in Fig. 1a (which also serves to show the typical vortices produced by fish when swimming). A net force (of the drag and thrust of the robotic fish body in the front-back direction) test in the flow tank indicates the shed vortex should be similar to those free swimming cases, otherwise, fish will not get the same thrust for the same kinematics.

R3.Q11:

7) I am not convinced by the experiments with real fish because as mentioned in the previous comment, the mechanism might be something

other than vortex interaction. In fact, the relation might be true but for other reasons than vortex interaction.

R3.A11:

We think this misunderstanding is due to the possible misinterpretation of the vortex-body interaction (vs. vortex-vortex hydrodynamic interactions) as pointed out above (R3.A4 and R3.A9). We apologise for that ambiguity. Indeed, as we think the Reviewer intuits, the mechanism we reveal in our robotic fish experiments, and on which our testable predictions and subsequent experimental validation is not vortex-vortex interactions, but vortex-body interactions.

REVIEWER COMMENTS

Reviewer #1 (Remarks to the Author):

The authors have done a commendable job addressing my many (maybe too many!) comments, and the revised paper is very much improved. I'll keep this follow-up report brief.

My remaining major gripe regards the Major Point #5 from my first report. I felt that it is not at all clear if the actual fish display phase locking that is quantitatively similar to the phase locking that is found to be energetically favorable in the robotic fish system. I still feel so, and I did not find that the authors' response cleared this up for me. This is a question that is at the heart of the work, touching on the basic interpretation that the fish are adopting a phase relationship that saves energy. Specifically, I'm looking at Fig. 2a, which tells us what phase relationships save energy and which cost energy. I'd like to put the phase relationship measured for fish on this same plot to see if they save energy. Using the data of Fig. 3 and the information in its caption, I see the "initial phase" is -0.2π . When I (mentally) plot this on Fig. 2a, I conclude that fish lie almost exactly along the red stripe, meaning they are consuming extra energy rather than saving energy! Am I missing something here? How can we conclude that fish are phase locking specifically to save energy?

From what I can tell from the presented data, I would conclude that fish indeed phase lock to vortices, but this may be quite unrelated to saving energy, since they seem to be at a high-cost phase. Not to keep pushing my own interpretation, but my intuition is that the fish are passively "forced" into such phase relationships due to hydrodynamic interactions. Is this conclusion any more or less justified than the energy-savings hypothesis? Why or why not?

Given that this point is so central to the paper, I'd like to see it addressed with abundant clarity, including changes in the manuscript. I'm especially concerned about sentences such as this from the Abstract: "Experiments with pairs of freely-swimming goldfish (*Carassius auratus*) reveal that, as predicted if employing VPM to save energy, fish typically adjust their relative tailbeat phase difference linearly as a function of front-back distance, allowing them to exploit neighbour-generated vortices." The wording here seems a little vague and potentially misleading. It is not enough to see any value of spatial phase and thus conclude energy savings. It is the particular numerical value of phase that is important, and adopting other phase values will actually cost extra energy. Do fish adopt the phase that saves energy in the robot experiments or costs extra energy?

Reviewer #3 (Remarks to the Author):

The authors have revised their manuscript and have better clarified what they mean by vortex phase matching. They have mentioned that they are referring to vortex-body interaction rather than the vortex-vortex interaction. However, constructive and destructive vortex-body interaction is still not clear to me. Looking at figure 2 and supplementary 6, here are my observations:

- 1) The phase difference is based on the body waves of the propulsors and not the actual vortex. In fact, it is assumed that the phase of the vortex is the same as the phase of the body wave. Therefore, the term vortex phase matching (VPM) is misleading. Why not use the term body phase matching or something of that sort? Such naming will help avoid the confusion that I had for other readers.

- 2) I still do not understand what constructive phase matching means from the fluid mechanics point of view. I understand from Fig. 2a,b that power consumption reduces for the follower at some phase angles, but what does constructive body-vortex interaction means? Is it possible just

by looking at the sketches of Fig. 2c say which one is constructive and which one is destructive without reference to the efficiency/cost values? I could not.

3) I am still not convinced that authors have looked at other possible mechanisms carefully, thereby vortex matching mechanism is still not well justified. I am not arguing that at certain body phase/distance the efficiency increases, but the reason for this increase can be something other than vortex interaction. As an example, here are two alternative mechanisms:

a) the leader's wake create a beneficial angle of attack for the follower's tail which increases the thrust production by the tail. It is known that main thrust is produced by the tail and maintaining an optimal angle of attack during the cycle can increase the thrust.

b) The fluid between the swimmers' bodies is pushed more efficiently at certain phase differences by the body undulations.

Specific comments:

Line 288: "constructive interactions between shed vortices,...": do you mean body-vortex? Clarify.

Line 411: "fish propulsion via muscle is far more efficient than by motors": that is debatable. The efficiency calculated for fish based on elongated body theory is known to be overestimated. Recent CFD studies calculate the efficiency of fish swimming at lower than 50%.

Reviewer #1 (Remarks to the Author):

R1.Q1:

The authors have done a commendable job addressing my many (maybe too many!) comments, and the revised paper is very much improved. I'll keep this follow-up report brief.

My remaining major gripe regards the Major Point #5 from my first report. I felt that it is not at all clear if the actual fish display phase locking that is quantitatively similar to the phase locking that is found to be energetically favorable in the robotic fish system. I still feel so, and I did not find that the authors' response cleared this up for me. This is a question that is at the heart of the work, touching on the basic interpretation that the fish are adopting a phase relationship that saves energy. Specifically, I'm looking at Fig. 2a, which tells us what phase relationships save energy and which cost energy. I'd like to put the phase relationship measured for fish on this same plot to see if they save energy. Using the data of Fig. 3 and the information in its caption, I see the "initial phase" is -0.2π . When I (mentally) plot this on Fig. 2a, I conclude that fish lie almost exactly along the red stripe, meaning they are consuming extra energy rather than saving energy! Am I missing something here? How can we conclude that fish are phase locking specifically to save energy?

R1.A1:

Thank you for making this important point — we agree. We have revised the title, abstract and several parts in the main text to address this issue. This has been extremely helpful since it allows us to convey the main conclusions more clearly. We have explained why we cannot directly compare the robotic fish and real fish data, due to inherent differences between them, and also we separate our analysis of VPM in the real fish from our analysis of body kinematics, since the former demonstrates that real fish use VPM and the latter indicates why they may do so. Since VPM has broad utility — for example it could be used both to increase thrust or to decrease energetic costs — we have made more clear that this is the main focus of our work. We have also changed the title of our paper to reflect better its interdisciplinary nature and its main focus.

The specific changes made are:

Title: We have changed it to "Vortex phase matching with a near-neighbour as a strategy for schooling in robots and in fish" to better reflect the main focus.

We have also changed the abstract to better differentiate from the analysis of VPM and the kinematic analysis that gives insight into why fish exhibit VPM.

Abstract: “...Experiments with pairs of freely-swimming goldfish (*Carassius auratus*) demonstrate that followers do exhibit a relative tailbeat phase difference that varies linearly as a function of front-back distance. Furthermore, we find that this behaviour---which is shown to require neither a functioning visual nor lateral line system---is consistent with the hypothesis that they use VPM typically, but not exclusively, to save energy.”

Line 314: “...Since Φ_0 also describes the phase difference at $D = 0$, we term this the initial phase difference. In practice, when estimating this value from our experimental data (be it from robots as here, or for real fish as below) we must estimate the relationship between Φ and D , employing the full range of D to best assess the phase difference, Φ , at $D=0$. This experimentally fitted Φ_0 is denoted as Φ_0^* . This is necessary for each experimental system due to inherent differences that exist between them, such as in body size, body morphology, body flexibility, differences in surface friction, and so on. If Eq. 2 with a specific Φ_0^* predicts phase difference Φ for a range of D , it indicates that the follower maintains a specific type of hydrodynamic interaction with the induced flow of the shed vortices across this range. However, the model alone can not tell if this specific interaction results in energy saving or not, since it depends on the value of Φ_0^* and, as noted above, on the details of the specific system. Thus the actual value of Φ_0^* for different systems (e.g. robotic and real fish) are not directly comparable. In the specific case of the robotic system, where the fitting can be made directly from values of energy consumption at different relative phases, Φ_0^* corresponds to maximum energy saving, and therefore $\Phi_0^*+\pi$ corresponds to maximum energetic costs.”

Line 396: “To test whether the observed linear relationship results from VPM (irrespective of why they may do so, which we will come to later), we employed Eq. 2, for each moment in time, to determine the phase difference that is predicted to occur---if the fish were employing VPM with a fixed Φ_0 ---across all front-back distances. We note that unlike for the robotic fish, we cannot conduct fitting directly to measured energetic costs, but can do so via measured body kinematics. Finding Φ_0^* in this case would indicate that, despite being a highly dynamic scenario---real fish constantly change their relative positions with respect to one another---they nonetheless adopt a consistent type of hydrodynamic interaction that is described by our model (Eq. 2, with the slope given directly from the measured quantities frequency f and swim speed u). By comparing the predicted and observed phase difference

over the full range of D using a periodic least square regression algorithm (Methods), we find the value of $\Phi_0^* = -0.2\pi$.”

Line 426: “While the above analysis demonstrates that fish are employing VPM, it is not possible by obtaining the Φ_0^* alone to determine why they are doing so. This is because Φ_0^* is fitted by estimating the typical phase difference at each front back distance, and as discussed above, Φ_0^* depends on system specific characteristics such as body morphology, body size, and so on. Unlike in the robotic fish experiments, where we can directly measure power consumption, such measurement is not possible for the real fish. We cannot directly apply the measured costs and benefits of swimming together from our bio-mimetic robots, due to inherent differences between them such as propulsive undulations travelling through the body of real fish (which is flexible and elastic) more smoothly than in our robotic fish (that has only three joints), and the skin of fish being coated with mucous to decrease resistance to fluids⁵¹ etc. We note that as a result of these factors real fish are likely to obtain considerably greater benefits if they employ VPM than do our robots.”

Line 440: “In order to gain insight into why fish perform VPM we conducted an additional analysis of the “power consumption” of the follower fish (by approximating it from the measured amplitude and frequency of its body undulation⁴⁸) for different types of hydrodynamic interactions (the full range of Φ_0 values, characterising the tail moving with, or against, the vortex induced flow for example)”

And in the SI:

Line 1348 : “In our model we assume that energy savings result from the hydrodynamic interactions between the fish body and the induced flow of the vortices shed by the leader. Without loss of generality, we simplify this as an effective point of the fish body that interacts with the flow. Based on this point’s phase we can define the relevant phase relationship between the leader and the follower, thus we get the initial phase difference Φ_0 . However, this effective point varies from system to system due to many factors such as fish body morphology, body kinematics and body friction. In the real fish data analysis we tracked the caudal peduncle since this represents well the body phase of the fish (Supplementary Movie 5). However this point may be slightly shifted from the effective point, resulting in a shift of the body phase. Therefore, the constant initial phase differences Φ_0^* may differ between systems, such as between our robotic and real fish.”

R1.Q2:

From what I can tell from the presented data, I would conclude that fish indeed phase lock to vortices, but this may be quite unrelated to saving energy, since they seem to be at a high-cost phase. Not to keep pushing my own interpretation, but my intuition is that the fish are passively “forced” into such phase relationships due to hydrodynamic interactions. Is this conclusion any more or less justified than the energy-savings hypothesis? Why or why not?

R1.A2:

Thank you for making this valuable point. We have made it much more explicit that while our analysis of VPM in real fish clearly shows that it does occur, that this analysis cannot, itself, inform us regarding why they do so. Since finding evidence of VPM is, in itself, important, we have now made a clear separation between that finding and our subsequent analysis of fish kinematics which shows behaviour consistent with the hypothesis that they do so, in part, to save energy. This also helps to prevent confusion that may arise by comparing directly the robotic fish and real fish data. As we have written in the response to the previous question we have changed the title and abstract to clarify this in addition to the detailed changes outlined in our response above. Our results suggest that “passive” response could be important (as indicated in the manuscript), but our kinematic analysis of the fish suggests that VPM in fish is associated with a behavioural response indicative that the fish are not typically in the high cost regime.

R1.Q3:

*Given that this point is so central to the paper, I'd like to see it addressed with abundant clarity, including changes in the manuscript. I'm especially concerned about sentences such as this from the Abstract: “Experiments with pairs of freely-swimming goldfish (*Carassius auratus*) reveal that, as predicted if employing VPM to save energy, fish typically adjust their relative tailbeat phase difference linearly as a function of front-back distance, allowing them to exploit neighbour-generated vortices.” The wording here seems a little vague and potentially misleading. It is not enough to see any value of spatial phase and thus conclude energy savings. It is the particular numerical value of phase that is important, and adopting other phase values will actually cost extra energy. Do fish adopt the phase that saves energy in the robot experiments or costs extra energy?*

R1.A3:

We agree and apologise for the unintentional lack of clarity. It has been very helpful to us to address this issue. As explained in **R1.A1**, above, we focus on the phenomenon and potential utility of vortex phase matching instead of energy saving

specifically. We also clarify that the finding of VPM does not necessarily imply it is employed for energy saving and that we cannot compare directly our robotic fish and real fish data. In addition, as written in our response **R1.A1**, we now make clear that determining Φ_0^* in itself does not imply energy saving behaviour, and that we cannot directly compare Φ_0^* between systems (it must be estimated from system-specific data). Thus possible energy saving behaviour in fish is now considered only when we analyse the fish kinematics to gain insight into whether they may be employing VPM to save energy.

R3.Q1:

The authors have revised their manuscript and have better clarified what they mean by vortex phase matching. They have mentioned that they are referring to vortex-body interaction rather than the vortex-vortex interaction. However, constructive and destructive vortex-body interaction is still not clear to me. Looking at figure 2 and supplementary 6, here are my observations:

1) The phase difference is based on the body waves of the propulsors and not the actual vortex. In fact, it is assumed that the phase of the vortex is the same as the phase of the body wave. Therefore, the term vortex phase matching (VPM) is misleading. Why not use the term body phase matching or something of that sort? Such naming will help avoid the confusion that I had for other readers.

R3.A1:

We agree that this important issue needed further clarification. We extensively discussed renaming the phenomenon but were unable to find a suitably concise term (for example we considered vortex body-phase matching, body-phase vortex matching, vortex-matching body undulation, vortex-induced-flow body-undulation matching). We also considered naming the observed phenomena “body phase matching” or something similar, but felt that this would imply that the bodies of the leader and the follower are phase matched, not capturing the important characteristic of the phenomena that it is dependent on front-back distance (for example, from Figs. 2, 3 and Supplementary Movie 5, the phase difference changes dynamically). Also the “phase of the vortex” shed by the leader (given by the direction of the induced flow at the locations) and the phase of the leader fish body are different (see Fig. 2d, e for examples). Therefore we now make very explicit our definition of VPM in the abstract to avoid any misunderstanding.

Line 28: “...This result is accounted for with a minimal model (informed by fluid visualisations) in which the follower adopts a tailbeat phase such that its undulating body

interacts in a consistent manner with the induced flow of the vortices shed downstream by the leader, which we term “vortex phase matching” (VPM).”

Fig. 2 Caption: “... Followers interact with the induced flow of vortices with the same body phase at any front-back distance (within the range of hydrodynamic interactions), termed vortex phase matching. ...”

We would, of course, be open to changing this term but we have failed to come up with a suitable alternative.

We also agree that the “constructive” and “destructive” may have been too strong/ misleading phrasing, so we revised this (please see more details in our answer at **R3.Q2**). This point is connected to the question raised by Reviewer1, so please also see our answer **R1.Q1**.

R3.Q2:

2) I still do not understand what constructive phase matching means from the fluid mechanics point of view. I understand from Fig. 2a,b that power consumption reduces for the follower at some phase angles, but what does constructive body-vortex interaction means? Is it possible just by looking at the sketches of Fig. 2c say which one is constructive and which one is destructive without reference to the efficiency/cost values? I could not.

R3.A2:

We agree that the constructive/destructive description was not clear enough. We removed these phrases throughout the manuscript and replaced them with the description of the fish tail movements along with (or against) the direction of the induced flow of the vortex shed by the leader. For example:

Line 285: “Visualisations of the shed vortices downstream of the robotic fish (estimated by visualising the motion of small hydrogen bubbles introduced into the flow (Supplementary Fig. 6, Movie 3)) reveal that in the energetically beneficial regions (indicated by the blue colour in Fig. 2a) the direction of the follower's tail during the moving coincides with the direction of the induced flow of the vortices shed by the leader (Fig. 2d and e, Supplementary Fig. 6, Movie 3).”

Line 533: “Comparing the kinematics of body undulations for given speeds in the presence, and absence, of neighbour-generated vortices indicates that fish tend to adjust their body undulations to interact with the induced flow of the vortices and that when they do so they

exhibit a lower tailbeat frequency and a higher tailbeat amplitude, behaviour that is consistent with fish exploiting vortices to save energy⁴⁸.”

Fig. 2 Caption: “...Energy cost is related to how the follower moves its body relative to the direction of the induced flow of the vortices, in the opposite direction with $\Phi_0 = \Phi_0^* + \pi$ (c) or in the same direction with $\Phi_0 = \Phi_0^*$ (d and e). Followers interact with the induced flow of vortices with the same body phase at any front-back distance (within the range of hydrodynamic interactions), termed vortex phase matching. (d and e; $\Phi_0 = \Phi_0^*$ describes the hydrodynamic interaction resulting in energy saving, see description in the text). As the front-back distance changes, the followers must dynamically adopt phase difference Φ , with respect to that of the leader.”

R3.Q3:

3) I am still not convinced that authors have looked at other possible mechanisms carefully, thereby vortex matching mechanism is still not well justified. I am not arguing that at certain body phase/distance the efficiency increases, but the reason for this increase can be something other than vortex interaction. As an example, here are two alternative mechanisms: a) the leader's wake create a beneficial angle of attack for the follower's tail which increases the thrust production by the tail. It is known that main thrust is produced by the tail and maintaining an optimal angle of attack during the cycle can increase the thrust. b) The fluid between the swimmers' bodies is pushed more efficiency at certain phase differences by the body undulations.

R3.A3:

As described in **R3.A2**, we have clarified our description of the interaction specifying that the the flow visualisation indicates that the tail does not interact directly with the vortex, but that energy saving corresponds to tail movement in the direction of the induced flow of the vortices shed by the leader.

Line 285: “Visualisations of the shed vortices downstream of the robotic fish (estimated by visualising the motion of small hydrogen bubbles introduced into the flow (Supplementary Fig. 6, Movie 3)) reveal that in the energetically beneficial regions (indicated by the blue colour in Fig. 2 a) the direction of the follower's tail during the moving coincides with the direction of the induced flow of the vortices shed by the leader (Fig. 2d and e, Supplementary Fig. 6, Movie 3).”

We apologise for this confusion, and we agree that the possible mechanisms mentioned by the reviewer could play a role, and we note that such mechanisms are not mutually exclusive. Informed by our robotic experiments we find that interacting with the induced flow of the vortices is an important factor, but the functional form of this relationship, as considered in our theoretical model, allows other mechanisms to play a role; the key feature is that phase difference exhibits a dependency on the front-back distance. While our model accounts for the robotic and real fish data it is true that additional mechanisms may play a role. To emphasise this point we have included the following texts in the manuscript:

Line 342: “As it remains possible that other swimming gaits and flow regimes may allow for different energy saving mechanisms^{3,10,42,43}, we refer to this specific relationship, and the associated mechanism, as vortex phase matching, or VPM.”

And Line 522: “Despite this good fit, there may of course be additional hydrodynamic effects, such as the channelling effect^{3,18} and the suction effect induced by low-pressure regions⁴³.”

And in the SI:

Line 1305: “Unsurprisingly the deviation for real fish, which might come from other hydrodynamic effects, such as the channelling effect^{3,18} and the suction effect induced by low-pressure regions⁴³, is larger than that of artificial system.”

R3.Q4:

Line 288: “constructive interactions between shed vortices, ...”: do you mean body-vortex? Clarify.

R3.A4:

We have now clarified this:

Line287: “... in the energetically beneficial regions (indicated by the blue colour in Fig. 2 a) the direction of the follower's tail during the moving coincides with the direction of the induced flow of the vortices shed by the leader (Fig. 2 d and e, Supplementary Fig. 6, Movie 3)....”

R3.Q5:

Line 411: “fish propulsion via muscle is far more efficient than by motors”: that is debatable. The efficiency calculated for fish based on elongated body theory is

known to be overestimated. Recent CFD studies calculate the efficiency of fish swimming at lower than 50%.

R3.A5:

We have removed this sentence.

REVIEWERS' COMMENTS:

Reviewer #1 (Remarks to the Author):

I thank the authors for carefully addressing my last round of comments regarding the specific phase relationship adopted by fish and the implications for energy savings/costs. In their responses and revisions to the manuscript, the authors state that what is learned about energetics in the robot system cannot be used to assess the energetics in real fish.

On one hand, this leads to awkwardness in the paper, clouding the main purpose for carefully measuring the dependence of energetics on relative spacing/phase in the robot system. The reader's natural tendency – and presumably the authors' original intent and main goal of the research – is to compare the robot and fish systems, but the paper now indicates that this cannot be done. This raises the question of why the two systems are included together in the same paper. Further, in my opinion, the stated reasons undermining a comparison are vague. There are surely many differences in the systems, but these are the kinds of differences that always arise in comparing an animal behavior to mechanical/robotic systems or to models and simulations. And such comparisons are done often and with success in learning something useful about the biological system.

This issue permeates the paper, whose title mentions a strategy. A strategy towards what goal? The casual reader will assume this to be saving energy, since that is what is emphasized in the robotic system, but the text says this cannot be concluded. I guess I am left feeling that, because the phase data from fish and robots do not align on the message of energy savings, the authors have opted to conclude they cannot be compared. But if they had aligned, we would all certainly be happy to conclude that fish save energy! Are results being bent to fit a predetermined conclusion?

On the other hand, the paper is satisfactory in that it is technically correct on this issue: indeed, it is unclear to what level of detail we can compare the results from robots and fish. And it must be said, as I have in my earlier responses, that I feel the paper has many merits that far outweigh my gripes. I commend the authors for pursuing such a challenging, important and interesting problem, and the results should be a very useful contribution to the field.

Reviewer #3 (Remarks to the Author):

The authors have adequately responded to my comments. Just one minor suggestion: it will make it clearer if some arrows are added to Fig. 2 c-e to show the direction of tail motion.

Reviewer #1 (Remarks to the Author):

R1.Q1:

Reviewer #1 (Remarks to the Author):

I thank the authors for carefully addressing my last round of comments regarding the specific phase relationship adopted by fish and the implications for energy savings/ costs. In their responses and revisions to the manuscript, the authors state that what is learned about energetics in the robot system cannot be used to assess the energetics in real fish.

On one hand, this leads to awkwardness in the paper, clouding the main purpose for carefully measuring the dependence of energetics on relative spacing/phase in the robot system. The reader's natural tendency – and presumably the authors' original intent and main goal of the research – is to compare the robot and fish systems, but the paper now indicates that this cannot be done. This raises the question of why the two systems are included together in the same paper. Further, in my opinion, the stated reasons undermining a comparison are vague. There are surely many differences in the systems, but these are the kinds of differences that always arise in comparing an animal behavior to mechanical/robotic systems or to models and simulations. And such comparisons are done often and with success in learning something useful about the biological system.

This issue permeates the paper, whose title mentions a strategy. A strategy towards what goal? The casual reader will assume this to be saving energy, since that is what is emphasized in the robotic system, but the text says this cannot be concluded. I guess I am left feeling that, because the phase data from fish and robots do not align on the message of energy savings, the authors have opted to conclude they cannot be compared. But if they had aligned, we would all certainly be happy to conclude that fish save energy! Are results being bent to fit a predetermined conclusion?

On the other hand, the paper is satisfactory in that it is technically correct on this issue: indeed, it is unclear to what level of detail we can compare the results from robots and fish. And it must be said, as I have in my earlier responses, that I feel the paper has many merits that far outweigh my gripes. I commend the authors for pursuing such a challenging, important and interesting problem, and the results should be a very useful contribution to the field.

R1.A1:

We thank you for your very helpful comments and future study suggestions. Instead of only focusing on energy saving in our initial version, we now focus first on whether real fish do indeed exhibit vortex phase matching, as we predicted they would, bearing in mind the multiple benefits that may be accrued by doing so (one of which is energy saving). This rule (Equation 2 in the main text) has two predictions in general: the first prediction describes the linear correlation (the vortex phase matching that is the main message in our paper), the second prediction describes the function of this linear correlation. In our real fish study, it is clear that our model (itself inspired directly by analyses of the biomimetic robots) predicts well the vortex phase matching mechanism and thus had considerable utility. We think this is the main message of our work, and here we can be confident that this general strategy may be widespread and may have considerable value for future development of biomimetic fish-like robots as well as to establish more effectively the nature of hydrodynamic interactions among real fish.

Had we then observed a perfect match to our experimental data we may well have thought that this is evidence for energy saving in the real system, but while certainly convenient we would likely have been wrong in making this assumption since the characteristics that determine these features (e.g. surface properties such as scales and their mucus coating vs. rubber, as well as different forces exerted by musculature vs. motors) do differ, and thus an exact match is likely to have been coincidental. Thus concluding anything regarding energy savings from an exact match, while making for an easier story, would likely have been erroneous. In future, it may be possible to make robots that can provide a quantitative match to a specific species of fish, but this will always be extremely challenging. Nonetheless, as we have shown here, robotics can be used to generate informed hypotheses and we provide methodology that allows experimentalists to fit our theory to their data in a way that does allow inherent differences to be taken into account.

While vortex phase matching could (and is likely to) be employed in context-dependent ways by real fish, such as to exploit vortices to increase thrust (relative to swimming alone) when a fish needs to do so (such as when moving to the front of a group to increase the probability of encountering suspended food particles), kinematic analyses of followers' body posture, a well-established technique (e.g. Liao *et. al*, 2003 *Science*), does provide evidence that the fish in our experiments did so in such way as would typically save energy. Thus we present the data not at all to fit a predetermined conclusion but employing the most robust and quantitative methodology available at present to both show that fish clearly employ VPM, a rule

that can confer multiple possible benefits, but also that kinematic analyses suggest that in the case of our real fish they likely benefit energetically by doing so.

R3.Q1:

The authors have adequately responded to my comments. Just one minor suggestion: it will make it clearer if some arrows are added to Fig. 2 c-e to show the direction of tail motion.

R3.A1:

Thanks for the positive comment. We revised the image accordingly in the revised version.